# Mechanism of ASF1 engagement by CDAN1

Samantha F. Sedor [1] & Sichen Shao [1,2] ✉

Codanin-1 (CDAN1) is an essential and ubiquitous protein named after congenital dyserythropoietic anemia type I, an autosomal recessive disease that manifests from mutations in *CDAN1* or *CDIN1* (CDAN1 interacting nuclease 1). CDAN1 interacts with CDIN1 and the paralogous histone H3-H4 chaperones ASF1A (Anti-Silencing Function 1 A) and ASF1B. However, CDAN1 function remains unclear. Here, we analyze CDAN1 complexes using biochemistry, single-particle cryo-EM, and structural predictions. We find that CDAN1 dimerizes and assembles into cytosolic complexes with CDIN1 and multiple copies of ASF1A/B. One CDAN1 can engage two ASF1 through two B-domains commonly found in ASF1 binding partners and two helices that mimic histone H3 binding. We additionally show that ASF1A and ASF1B have different requirements for CDAN1 engagement. Our findings explain how CDAN1 sequesters ASF1A/B by occupying all functional binding sites known to facilitate histone chaperoning and provide molecular-level insights into this enigmatic complex.

Congenital dyserythropoietic anemias (CDAs) are inherited disorders primarily characterized by failure of the erythroid lineage to effectively differentiate into red blood cells. CDAs are classified into subtypes based on distinctive bone marrow morphologies arising from mutations in specific genes[1–4]. Some CDA-associated genes, such as *GATA1*[5] and *KLF1*[6], encode transcription factors that primarily function in hematopoiesis. Other CDA genes are linked to fundamental cellular functions. These include *SEC23B*[7,8], which encodes a component of the COPII coat that mediates anterograde vesicular transport, and *KIF23*[9] and *RACGAP1*[10], which encode subunits of the centralspindlin complex required for cytokinesis.

Most genes associated with CDAs are extensively studied. However, *CDAN1* and *CDIN1* (*C15orf41*), the two genes linked to CDA subtype 1 (CDA-I), encode a protein complex of unknown function[11–14]. A striking feature of CDA-I is the formation of spongy or "Swiss cheese" heterochromatin in erythroblast nuclei, a defect in chromatin compaction not seen in other CDAs[15–20]. Consistent with the possibility that this complex regulates chromatin condensation, CDAN1 is reported to interact with ASF1A and ASF1B[21], paralogous histone chaperones that we will refer to as ASF1 in interchangeable contexts throughout this study. ASF1 binds the H3-H4 dimer and shuttles between the cytosol and nucleus to facilitate downstream nucleosome assembly[22–27]. CDAN1 is proposed to inhibit this function by sequestering ASF1 in complex with H3-H4 in the cytosol via a 'B'-domain[21], a short linear

motif containing two consecutive basic residues. B-domains are employed by multiple ASF1 binding partners, including the downstream histone H3-H4 assembly chaperones HIRA[28–32]. CDIN1 contains a predicted PD-(D/E)XK nuclease domain[11], but no nucleic acid substrates have been conclusively linked to its putative enzymatic activity.

Despite their link to CDA-I, CDAN1 and CDIN1 probably mediate a fundamental cellular function that is not restricted to erythropoiesis. Both *CDAN1* and *CDIN1* are ubiquitously expressed and essential[33–35]. Notably, knocking out either gene is embryonic lethal in mice prior to the onset of red blood cell production[33,34]. In addition, basic questions regarding the molecular interactions of these proteins remain outstanding.

In this work, we integrate genetic, biochemical, and structural approaches to investigate the interactions of CDAN1 with its binding partners. We endogenously tag CDAN1 and CDIN1 to show that these proteins form an obligate cytosolic complex that simultaneously binds ASF1 but not histones. Biophysical measurements and single-particle cryogenic electron microscopy (cryo-EM) additionally reveal that CDAN1 dimerizes and can engage multiple ASF1 molecules through distinct B-domains and helices that mimic histone H3 binding to ASF1. Furthermore, we find that these elements engage ASF1A and ASF1B differently. These results elucidate the interactions used by CDAN1 to recruit and inhibit the chaperone function of ASF1, thus providing mechanistic insights into this essential complex.

[1]Department of Cell Biology, Harvard Medical School, Boston, MA, USA. [2]Howard Hughes Medical Institute, Boston, MA, USA.
✉e-mail: sichen_shao@hms.harvard.edu

## Results

### Endogenous CDAN1 and CDIN1 assemble together with ASF1

To investigate CDAN1 and CDIN1 in dividing cells without potential artifacts of overexpression, we used CRISPR-Cas9 methods to endogenously tag CDAN1 or CDIN1 in Flp-In 293 T-REx cells with a C-terminal HaloTag-FLAG (HF) epitope (Supplementary Fig. 1a). CDAN1 is reported to interact with CDIN1 and ASF1 through distinct binding sites[21,36,37], but it is unclear if CDAN1 can bind these factors at the same time. Consistent with prior studies, immunoprecipitations of endogenous CDAN1-HF recovered CDIN1 and both ASF1 paralogs (Fig. 1a, lane 9). Immunoprecipitations of endogenous CDIN1-HF also recovered CDAN1, as well as ASF1A and ASF1B (Fig. 1a, lane 8, and Supplementary Fig. 1b). In addition, endogenous CDIN1 was

significantly depleted from the flow-through of CDAN1-HF immunoprecipitations (Fig. 1a, lane 6) and vice versa (Fig. 1a, lane 5), indicating that CDAN1 and CDIN1 primarily exist in complex with each other. These findings suggest that CDAN1 can bind CDIN1 and ASF1 simultaneously and confirm that the HF tag does not disrupt the interactions between these factors.

Next, we used HaloPROTAC3 (HP3), a small molecule degrader for HaloTag fusion proteins[38,39], to acutely deplete CDAN1-HF or CDIN1-HF (Fig. 1b). We achieved robust and specific degradation of each HF-tagged protein upon treatment with HP3 (Fig. 1b, lanes 3 and 6) but not an inactive isomer (Ent-HP3) (Fig. 1b, lanes 2 and 5). Importantly, acute degradation of CDAN1-HF destabilized endogenous CDIN1 (Fig. 1b, lane 6). In contrast, degradation of CDIN1-HF did not change

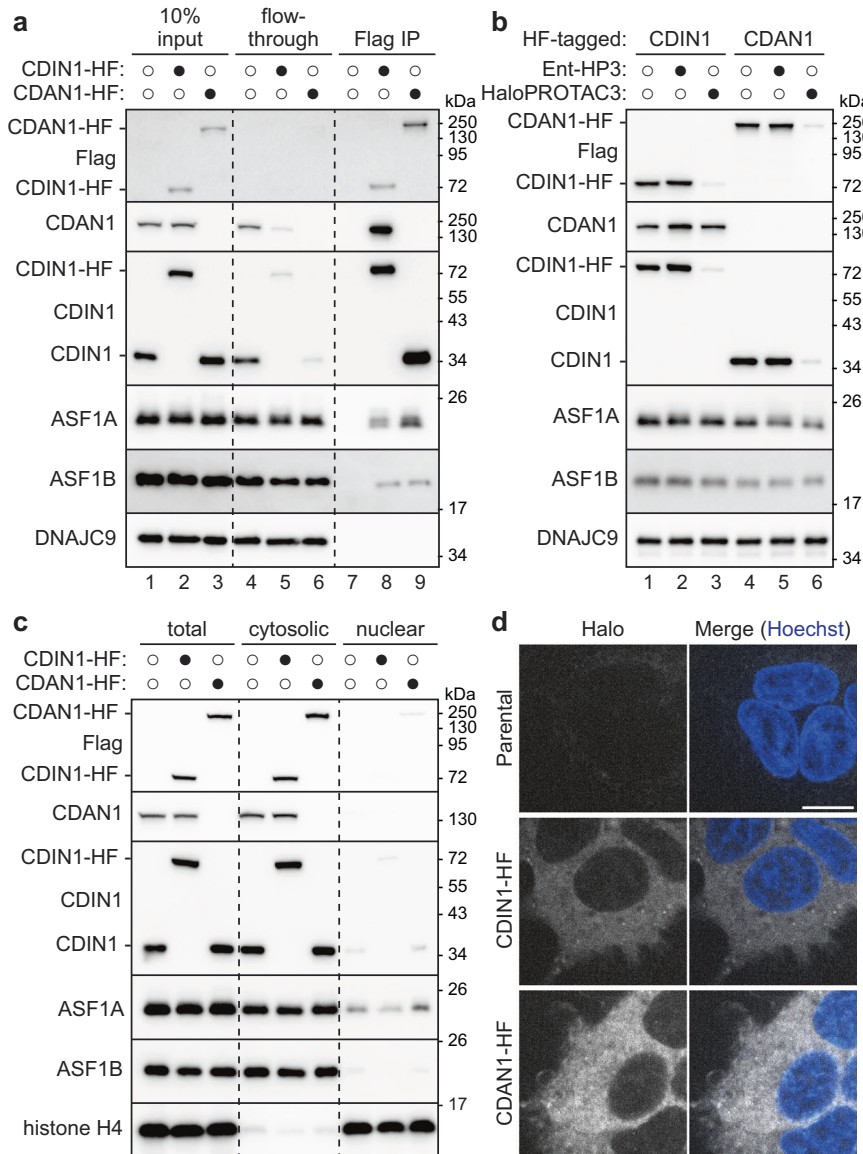

**Fig. 1 | CDAN1 and CDIN1 assemble together with ASF1 in the cytosol.**
**a** Endogenous CDAN1 and CDIN1 assemble with ASF1. Flp-In 293 T-REx cells without or with a C-terminal HaloTag-FLAG (HF) tag on endogenous CDIN1 or CDAN1 were lysed (input), subjected to anti-FLAG immunoprecipitations (IP), and analyzed by SDS-PAGE and immunoblotting; representative of 3 independent replicates. Note: the CDAN1 antibody raised against a C-terminal epitope does not recognize CDAN1-HF. Where applicable, different protein populations recognized by the same antibody are labeled. **b** CDIN1 stability requires CDAN1. Parental, CDIN1-HF or CDAN1-HF cells were treated without or with 500 nM HaloPROTAC3 (HP3) or an inactive

enantiomer (ent-HP3) for 72 hr, lysed, and analyzed by SDS-PAGE and immunoblotting; representative of at least 3 independent replicates. **c** CDAN1 and CDIN1 are cytosolic. Parental, CDIN1-HF, or CDAN1-HF cell lysates (total) were fractionated to separate cytosolic and nuclear contents and analyzed by SDS-PAGE and immunoblotting; representative of 3 independent replicates. **d** Live-cell images of parental, CDIN1-HF, and CDAN1-HF cells labeled with JFX650 HaloTag ligand (gray) and Hoechst (blue), representative of 3 independent replicates. Scale bar, 10 μm. Source Data are provided as a Source Data file.

endogenous CDAN1 levels (Fig. 1b, lane 3). We observed the same dependency of CDIN1 levels on CDAN1 using siRNA-mediated knockdowns[19] (Supplementary Fig. 1c), while ASF1A and ASF1B levels were not impacted by depletion of either protein. Thus, the stability of CDIN1 depends on CDAN1[36], supporting the interpretation that CDIN1 forms an obligate complex with CDAN1.

The interaction between CDAN1 and ASF1A is cytosolic[21]. However, several studies report nuclear localization of CDIN1 and/or CDAN1[19,20,36,37,40–42]. Given our results that CDAN1 can form a complex containing both ASF1 and CDIN1 and that most of CDIN1 appears to be bound to CDAN1, we examined the localization of these proteins using independent methods. Using our endogenously tagged cell lines, we performed biochemical fractionations followed by immunoblotting with validated antibodies (Fig. 1c), as well as live imaging after labeling the HaloTag fusion proteins with a fluorophore-conjugated ligand (Fig. 1d). Both approaches demonstrate that endogenous CDAN1 and CDIN1 are primarily cytosolic in Flp-In 293 T-REx cells (Fig. 1c, d). This result was initially unexpected, considering prior immunofluorescence results using CDIN1 antibodies[19,20,40]. We confirmed that for the antibody previously used in HEK293 cells[40], the immunofluorescence signal did not decrease upon siRNA-mediated knockdown of CDIN1 (Supplementary Fig. 1c, d) or HP3-mediated degradation of CDIN1-HF (Supplementary Fig. 1e, f), suggesting that the antibody is nonspecific[37]. Our data, therefore, show that endogenous CDAN1 assembles into a cytosolic complex with CDIN1 and ASF1 paralogs.

## CDAN1 complexes contain multiple copies of each subunit
Next, using transiently transfected Expi293 cells, we performed tandem purifications of Strep-tagged CDAN1 (ST-CDAN1) in complex with FLAG-tagged CDIN1 (F-CDIN1) and HA-tagged ASF1A (Fig. 2a, C:C:A complex for CDAN1:CDIN1:ASF1A), or with only FLAG-tagged ASF1A (F-ASF1A) (Fig. 2b, C:A complex for CDAN1:ASF1A). Immunoblotting of samples collected at different steps of the C:C:A and C:A purification procedures revealed several notable insights. First, immunoprecipitation of F-CDIN1 substantially depleted ST-CDAN1 from the lysate (Fig. 2a, compare lanes 2 and 3), indicating that these two proteins are usually associated with each other, even when overexpressed. Second, both purifications contained low but detectable amounts of endogenous CDIN1 and ASF1A, even if their tagged counterpart was used for purification (Fig. 2a, b, lanes 4 and 6). This observation suggests that multiple copies of each factor are present in the complexes. Finally, immunoprecipitation of F-ASF1A recovered a detectable level of histone H3 (Fig. 2b, lane 4) that was lost after subsequently pulling down on ST-CDAN1 (Fig. 2b, lane 6). H3 also was not observed after F-CDIN1 immunoprecipitation (Fig. 2a, lane 4). This suggests that CDAN1 complexes do not sequester the histone H3-H4 dimer with ASF1 in the cytosol, as previously suggested[21].

The C:C:A and C:A complexes migrated as single peaks by size exclusion chromatography (SEC) (Fig. 2c and Supplementary Fig. 2a), indicating stable assembly of the subunits, and negative stain EM images of the C:C:A complex showed discrete particles (Supplementary Fig. 2b). Both the C:C:A and the C:A complexes also migrated at a larger size than expected for an assembly containing one copy of each subunit[37]. To determine the molecular weights of these complexes more precisely, we performed SEC with multi-angle light scattering (SEC-MALS) and found that the measured molar mass of the recombinant C:A complex, 364.3 kDa, most closely aligns with a stoichiometry of two CDAN1 and three ASF1A (Fig. 2c and Supplementary Fig. 2c). Similarly, the measured molar mass of the C:C:A complex, 432.1 kDa, is most consistent with a stoichiometry of two CDAN1, two CDIN1, and three ASF1A. In addition, the difference between the measured molar mass of the C:C:A and C:A complexes matches the molecular weight of two CDIN1. These data suggest that CDAN1 dimerizes and is capable of binding multiple ASF1A molecules despite containing only one described B-domain[21].

## Cryo-EM structures of CDAN1 complexes
To understand the interactions that mediate CDAN1 dimerization and ASF1A association, we determined single-particle cryo-EM structures of the C:C:A complex (Fig. 3, Supplementary Figs. 3 and 4, Table 1, and Supplementary Movie 1). 3D classification strategies allowed us to obtain cryo-EM maps at overall resolutions ranging from 3.0 to 3.5 Å of a CDAN1 dimer with either two or three ASF1A molecules clearly resolved (Fig. 3b, c and Supplementary Figs. 3 and 4).

Based on homology searches, CDAN1 contains two MIF4G (middle domain of eukaryotic initiation factor 4G) domains (MIF4G1, MIF4G2), a three-helix bundle of unknown function (DUF3819), and an MA3 domain commonly found in MIF4G proteins[43–45] (Fig. 3a and Supplementary Fig. 5a–d). This domain architecture, particularly the distinctive presence of the three-helix bundle, resembles the central portion of the CNOT1 protein that serves as a scaffold in the CCR4-NOT deadenylation complex[36,46–49] (Supplementary Fig. 5c–f). With our cryo-EM maps, we could model both MIF4G domains of CDAN1 (Fig. 3). This showed that MIF4G2, including a conserved central extension (residues 645–675), mediates dimerization (Fig. 3 and Supplementary Fig. 5a). MIF4G1 is split by a long loop (residues 64–298) that contains the B-domain previously identified to interact with ASF1[21] (Fig. 3 and Supplementary Fig. 5a). Both this loop and features extending from MIF4G2 contain additional elements that facilitate ASF1A interaction, discussed below.

Although present in the purified complex (Fig. 2a), CDIN1 and the C-terminal portion of CDAN1 (residues 833-1227, including the three-helix bundle and the MA3 domain that interacts with CDIN1[36,37,50]) were not visible in our cryo-EM maps. This suggests that the C-terminus of CDAN1 is flexible relative to the N-terminal core. To obtain experimental insights into this interaction, we purified the CDAN1 MA3 domain (ST-CDAN1$_C$, residues 1008-1227) in complex with CDIN1 (Supplementary Fig. 6a, b). Single-particle cryo-EM analysis of this complex resulted in a ~ 6 Å reconstruction consistent with the extensive CDAN1:CDIN1 interface predicted by Alphafold3[51] (Supplementary Fig. 6c–f) and recent size exclusion chromatography coupled with small angle X-ray scattering (SEC-SAXS) analysis[50]. These results support a role for CDAN1 as a scaffold that orchestrates the assembly of CDIN1 and multiple copies of ASF1 to distinct regions of the same complex.

## CDAN1 interacts with ASF1A through distinct B-domains
Surprisingly, even though CDAN1 homodimerizes, we consistently observed asymmetric assemblies during 2D and 3D classifications of our cryo-EM data (Fig. 3b, Supplementary Fig. 3, and Supplementary Movie 1). The asymmetry is explained by the association of two ASF1A molecules with one CDAN1 monomer and a single ASF1A with the other (Fig. 3b and Supplementary Fig. 3). Notably, symmetry expansion along the C2 dimer axis followed by focused classification revealed an approximately 50:50 split of CDAN1 monomers engaged with one or two ASF1A (Supplementary Fig. 3), consistent with an asymmetric complex. Although there is evidence for some conformational heterogeneity within our dataset (Supplementary Fig. 3 and Supplementary Movie 1), this architecture was the most prominent reconstruction, and other putative conformations did not resolve at higher resolutions. In addition, this stoichiometry matches our SEC-MALS measurements supporting the predominant presence of three ASF1A in both the C:A and C:C:A complexes (Fig. 2c). In our cryo-EM maps, the two ASF1A molecules on one side of the complex appear to stack on top of each other on a platform formed primarily by the MIF4G2 domain of CDAN1. We refer to the ASF1A proximal to the CDAN1 core as ASF1A-1 and the distal ASF1A as ASF1A-2 (Fig. 3b–d).

To better visualize the interactions that mediate this configuration of ASF1A-1 and ASF1A-2, we performed an intermediate 3D refinement with C2 symmetry imposed, which we then used for a symmetry expansion and 3D classification of the expanded particle set with masks focused on only one side of the dimeric complex

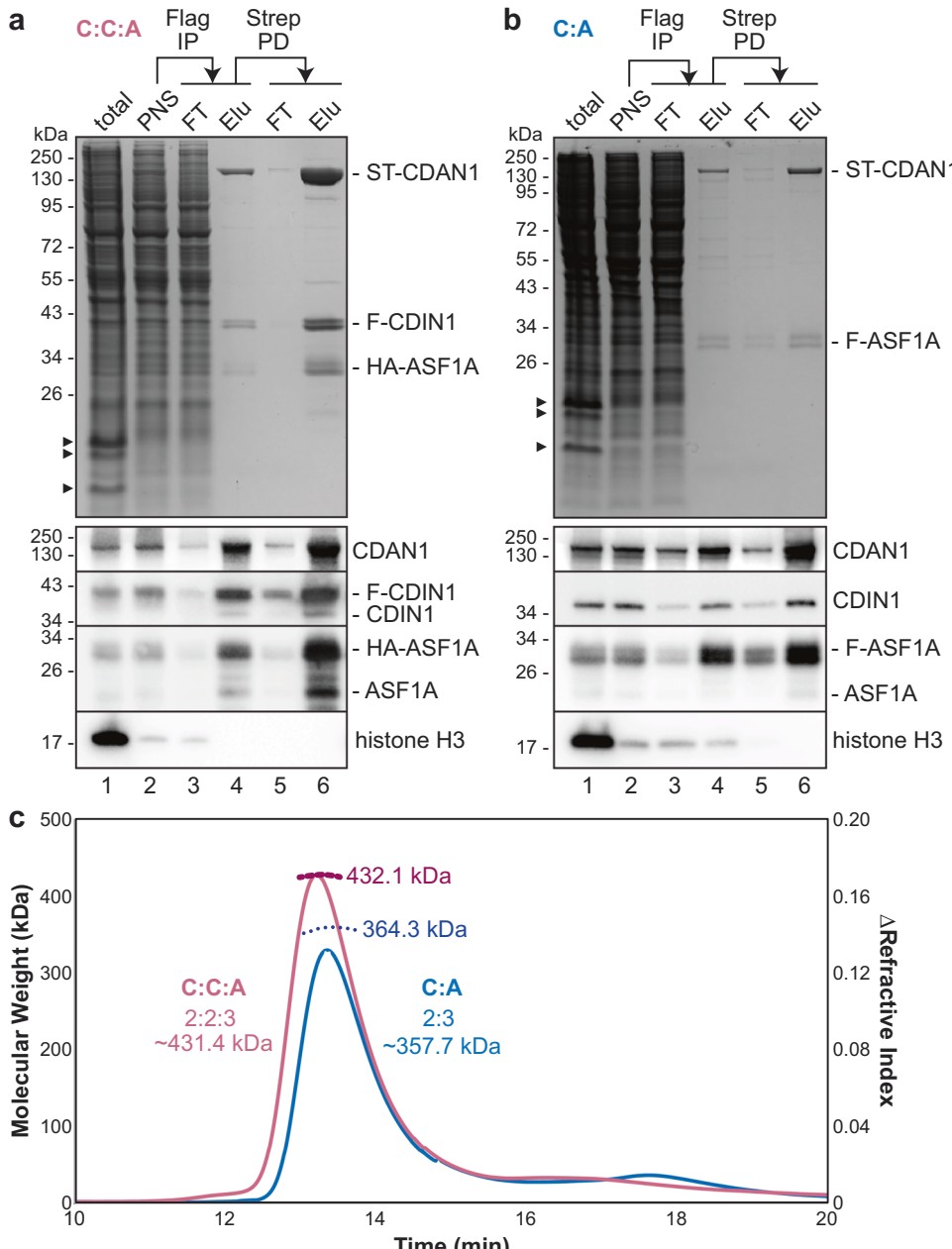

**Fig. 2 | CDAN1 complexes contain multiple copies of each subunit. a** Strep-tagged CDAN1 (ST-CDAN1), FLAG-tagged CDIN1 (F-CDIN1), and HA-tagged ASF1A were overexpressed in Expi293 cells by transient transfection for 72 hr. The cells were then lysed, and the post-nuclear supernatants (PNS) were subjected to anti-FLAG immunoprecipitation (FLAG IP) followed by Strep-Tactin pulldowns (Strep PD) to purify the CDAN1-CDIN1-ASF1A (C:C:A) complex. The input, flow-through (FT), and elution (Elu.) samples were analyzed by SDS-PAGE and Coomassie staining (top) or immunoblotting (bottom); representative of 3 independent replicates.

Arrowheads denote histones. **b** As in (**a**), except with overexpression of ST-CDAN1 and FLAG-tagged ASF1A (F-ASF1A) to purify the CDAN1-ASF1A (C:A) complex. **c** Size exclusion chromatography with multi-angle light scattering (SEC-MALS) of the C:C:A (pink) and C:A (blue) complexes determined molecular weights consistent with 2 copies of CDAN1, 3 copies of ASF1A, and, in the C:C:A complex, 2 copies of CDIN1. The predicted molecular weights of these stoichiometries are indicated. Source data are provided as a Source Data file.

(Supplementary Fig. 3). This specifically isolated classes with clear occupancy of two ASF1A that generated a 3.2 Å reconstruction after focused refinement (Fig. 3c and Supplementary Fig. 3). These maps all showed two distinct CDAN1 B-domain densities, one interacting canonically with each ASF1A, that we refer to as $BD_{A1}$ and $BD_{A2}$ (Figs. 3b–d, 4a–d, and Supplementary Fig. 4d).

Only one B-domain sequence (residues 193-203), which resides in a long loop extending from the MIF4G1 domain of CDAN1, has been identified previously[21] (Fig. 3a). Careful inspection of the CDAN1 sequence revealed another conserved segment (residues 823–832)

consistent with a B-domain (Fig. 4d). We assigned this sequence to $BD_{A1}$ because it extends directly from structured regions of CDAN1, ending in residue 818, adjacent to the $BD_{A1}$ density that interacts with ASF1A-1 (Fig. 4b and Supplementary Fig. 7a). Notably, the linker between this B-domain sequence and the structured CDAN1 core is not long enough to reach the B-domain interaction with the distal ASF1A-2 (Fig. 3a, c and Supplementary Fig. 7a). In this placement, $BD_{A1}$ makes canonical interactions with ASF1A-1, where the consecutive basic residues R825 and K826 interact with D58 and D37 of ASF1A-1, respectively (Fig. 4a, b).

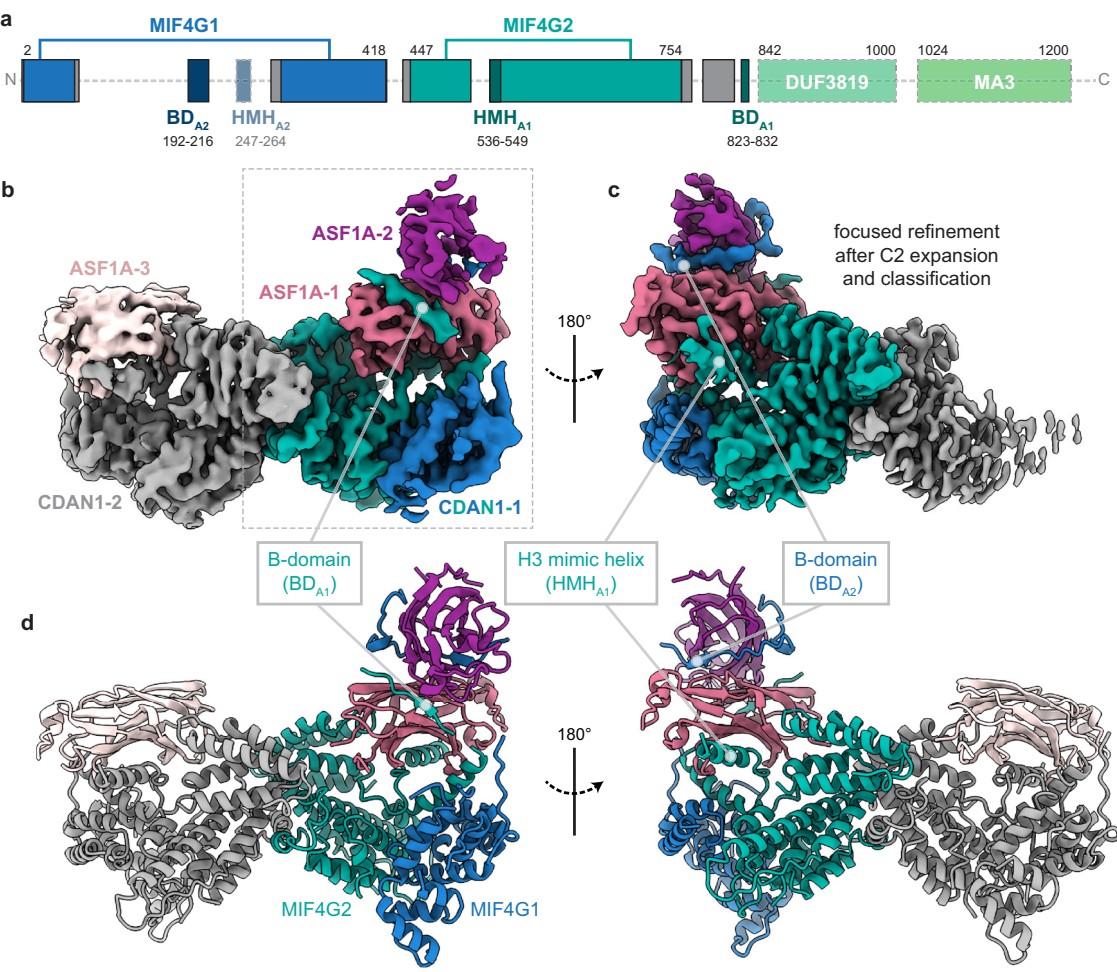

**Fig. 3 | Cryo-EM structures of CDAN1-ASF1A complexes. a** Domain scheme of CDAN1. Gray boxes indicate structured regions modeled based on the cryo-EM maps that fall outside of defined domains. Unresolved regions are indicated by transparent boxes for regions predicted to be structured or a dashed line for unstructured regions. Domains are colored as follows: MIF4G1, blue, MIF4G2, teal, DUF3819, sea green, and MA3, green. The ASF1 binding elements $BD_{A1}$ (residues 823–832), $BD_{A2}$ (residues 192–216 are modeled, which include the core B-domain residues 193–203 and conserved C-terminal extension residues 204–216), $HMH_{A1}$ (residues 247–264), and $HMH_{A2}$ (unmodeled residues 247–264) are indicated. **b**, **c** Cryo-EM maps of the C:C:A complex showing (**b**), three ASF1A molecules bound to a CDAN1 dimer (contoured at 9.4σ) or (**c**) focused on CDAN1 interactions with two stacked ASF1 molecules (contoured at 5.8σ) after C2 symmetry expansion and 3D classification focused on one arm of the complex. **d** Model of a CDAN1 dimer bound to three ASF1A. All colors in (**b–d**) correspond to the domain map in (**a**).

We assigned the previously identified B-domain sequence as $BD_{A2}$ that interacts canonically with ASF1A-2 (Fig. 4c), where the consecutive basic residues R195 and R196 interact with D58 and D37 of ASF1A-2, respectively. In addition, the loop containing $BD_{A2}$ extends into additional conserved residues (204–216) (Figs. 3a and 4d). Alphafold3[51] predictions of a CDAN1 monomer with two copies of ASF1A indicate that this conserved segment wraps around the ASF1A that binds $BD_{A2}$ (Supplementary Fig. 7b, c). This interaction is consistent with density in our cryo-EM maps (Fig. 3b and Supplementary Figs. 4d and 7a) and may help stabilize the position of ASF1A-2 stacked on top of ASF1A-1. Hence, we have identified a second B-domain in CDAN1 that facilitates the recruitment of two ASF1A molecules.

**CDAN1 has ASF1 client mimicry elements**
In addition to $BD_{A1}$, CDAN1 also interacts with ASF1A-1 through a helix (residues 536–549) that binds the same interface ASF1 uses to engage a helix of histone H3[22,52,53] (Fig. 4d, f and Supplementary Fig. 7d). We refer to this H3 mimic helix as $HMH_{A1}$. Like H3 (Fig. 4e), $HMH_{A1}$ of CDAN1 makes hydrophobic interactions with L96 and V94 of ASF1A-1 through L549 and L545, as well as electrostatic interactions with D54 and D88 of ASF1A-1 through R541 and R548 (Fig. 4f). This client mimicry mode of

ASF1 recruitment is also employed by TLK2, a kinase that phosphorylates ASF1A and ASF1B[54].

Our assignments of $HMH_{A1}$ and the two B-domains described above are consistent with Alphafold3 predictions of CDAN1 with two ASF1A, including the observation that $BD_{A1}$ and $HMH_{A1}$ bind to the same ASF1A molecule (Supplementary Fig. 7b). Notably, Alphafold3 also predicts that another CDAN1 helix (residues 247–264), which resides in the same loop that contains $BD_{A2}$ (Fig. 3a), binds the H3-binding interface of the other ASF1A molecule associated with $BD_{A2}$ (Supplementary Fig. 7b, c). The placement of this helix is consistent with density at the equivalent interface of ASF1A-2 visible in unsharpened cryo-EM maps of the C:C:A complex (Supplementary Fig. 7c). We therefore refer to this helix as $HMH_{A2}$ (Fig. 4d, g and Supplementary Fig. 7b–d). Compared to the H3 helix and $HMH_{A1}$, the predicted N-to-C orientation of $HMH_{A2}$ is reversed yet still positions conserved amino acids to make key interactions with the H3-binding interface of ASF1A (Fig. 4d, g and Supplementary Fig. 7d). In this position, L247 and L254 of $HMH_{A2}$ would satisfy hydrophobic interactions with L96 and V94 of ASF1A-2, and R251 and R258 of $HMH_{A2}$ would interact with D88 and D54 of ASF1A-2 (Fig. 4g).

CDAN1 interactions with ASF1A-1 and ASF1A-2 occupy not only the site of H3 binding but also potentially the site of interaction between

**Table 1 | Cryo-EM data collection, refinement, and validation**

|  | CDAN1 + 3xASF1A<br>EMD-45959<br>PDB 9CVC | CDAN1 + 2xASF1A (C1)<br>EMD-45960 | CDAN1 + 2xASF1A<br>(C2 expanded & focused)<br>EMD-45961 |
|---|---|---|---|
| **Data collection and processing** | | | |
| Magnification | 105,000 | 105,000 | 105,000 |
| Voltage (kV) | 300 | 300 | 300 |
| Electron exposure (e–/Å$^2$) | 50.3 | 50.3 | 50.3 |
| Defocus range (µm) | 1.3–2.3 | 1.3–2.3 | 1.3–2.3 |
| Pixel size (Å) | 0.825 | 0.825 | 0.825 |
| Symmetry imposed | C1 | C1 | C1 |
| Initial particle images (no.) | 1,665,601 | 1,885,342 | 1,885,342 |
| Final particle images (no.) | 25,056 | 256,296 | 122,061 |
| Map resolution (Å) | 3.5 | 3.0 | 3.2 |
| FSC threshold | 0.143 | 0.143 | 0.143 |
| Map resolution range (Å) | 3.1–50.5 | 2.6–6.6 | 2.6–5.9 |
| **Refinement** | | | |
| Initial model used | Alphafold | – | – |
| Model resolution (Å) | 4.0 | – | – |
| FSC threshold | 0.5 | – | – |
| Model resolution range (Å) | 4.0 to 45.1 | – | – |
| Map sharpening $B$ factor (Å$^2$) | – 53.7 | – | – |
| Model composition | | | |
| Non-hydrogen atoms | 12,564 | – | – |
| Protein residues | 1572 | – | – |
| $B$ factors (Å$^2$) | | | |
| Protein | 106.91 | – | – |
| R.m.s. deviations | | | |
| Bond lengths (Å) | 0.005 | – | – |
| Bond angles (°) | 1.038 | – | – |
| Validation | | | |
| MolProbity score | 2.14 | – | – |
| Clashscore | 14.29 | – | – |
| Poor rotamers (%) | 0 | – | – |
| Ramachandran plot | | | |
| Favored (%) | 92.33 | – | – |
| Allowed (%) | 7.41 | – | – |
| Disallowed (%) | 0.26 | – | – |

ASF1A and the C-terminal tail of histone H4[53] (Supplementary Fig. 7d). Specifically, a loop (residues 468–476) within the CDAN1 MIF4G2 domain would clash with an H4 tail positioned on ASF1A-1, and the BD$_{A2}$ extension would clash with an H4 tail positioned on ASF1A-2 (Supplementary Fig. 7d). Altogether, our structural analyses show that a single CDAN1 molecule can recruit two ASF1A molecules through two distinct B-domains and two H3 mimic helices. These interactions occupy all functional binding interfaces of both ASF1A proteins known to engage histones, downstream chaperones such as HIRA, and regulators such as TLK2. These findings indicate that ASF1A does not perform its canonical histone chaperoning activity when bound to CDAN1.

**All CDAN1 binding elements contribute to ASF1 recruitment**

We next used mutagenesis to validate the ASF1 binding elements of CDAN1. For each element (HMH$_{A1}$, BD$_{A1}$, HMH$_{A2}$, and BD$_{A2}$), we mutated three key residues implicated in ASF1 binding to alanines (Fig. 5a). We then tested each of these mutant binding elements (HMH$_{A1}$*, BD$_{A1}$*, HMH$_{A2}$*, and BD$_{A2}$*) individually and in combination to assess their impact on the copurification of endogenous ASF1A, ASF1B, and CDIN1 with ST-CDAN1 (Fig. 5a).

Interestingly, pulldowns of these ST-CDAN1 variants revealed that ASF1B was more dependent than ASF1A on HMH$_{A1}$ for interaction with CDAN1 (Fig. 5a). While wildtype (WT) CDAN1 pulled down both ASF1A and ASF1B, as expected (Fig. 5a, lane 1), HMH$_{A1}$* CDAN1 copurified ASF1A but not ASF1B (Fig. 5a, lane 2). The interaction of ASF1B with CDAN1 was also more sensitive to mutation of BD$_{A1}$ (Fig. 5a, lane 3). Consistent with these results, HMH$_{A1}$* BD$_{A1}$* ST-CDAN1 harboring mutations in both elements that bind the proximal ASF1 (Fig. 4b, f) pulled down ASF1A but not ASF1B (Fig. 5a, lane 8). In contrast, HMH$_{A2}$* BD$_{A2}$* ST-CDAN1 harboring mutations in both elements that bind the distal ASF1 still pulled down both ASF1A and ASF1B (Fig. 5a, lane 9). Thus, ASF1B, but not ASF1A, is nearly exclusively reliant on the proximal HMH for stable binding to CDAN1.

Mutating both B-domains (Fig. 5a, lane 6) or both H3 mimic helices (Fig. 5a, lane 7) abrogated the binding of both ASF1 paralogs. Mutating HMH$_{A1}$ in combination with BD$_{A2}$ also abrogated the binding of ASF1A and ASF1B (Fig. 5a, lane 10). However, the reciprocal double mutant, BD$_{A1}$* HMH$_{A2}$* ST-CDAN1, still pulled down both ASF1 paralogs (Fig. 5a, lane 11). Mutating all four binding elements or three in any combination also impaired the copurification of ASF1 (Supplementary

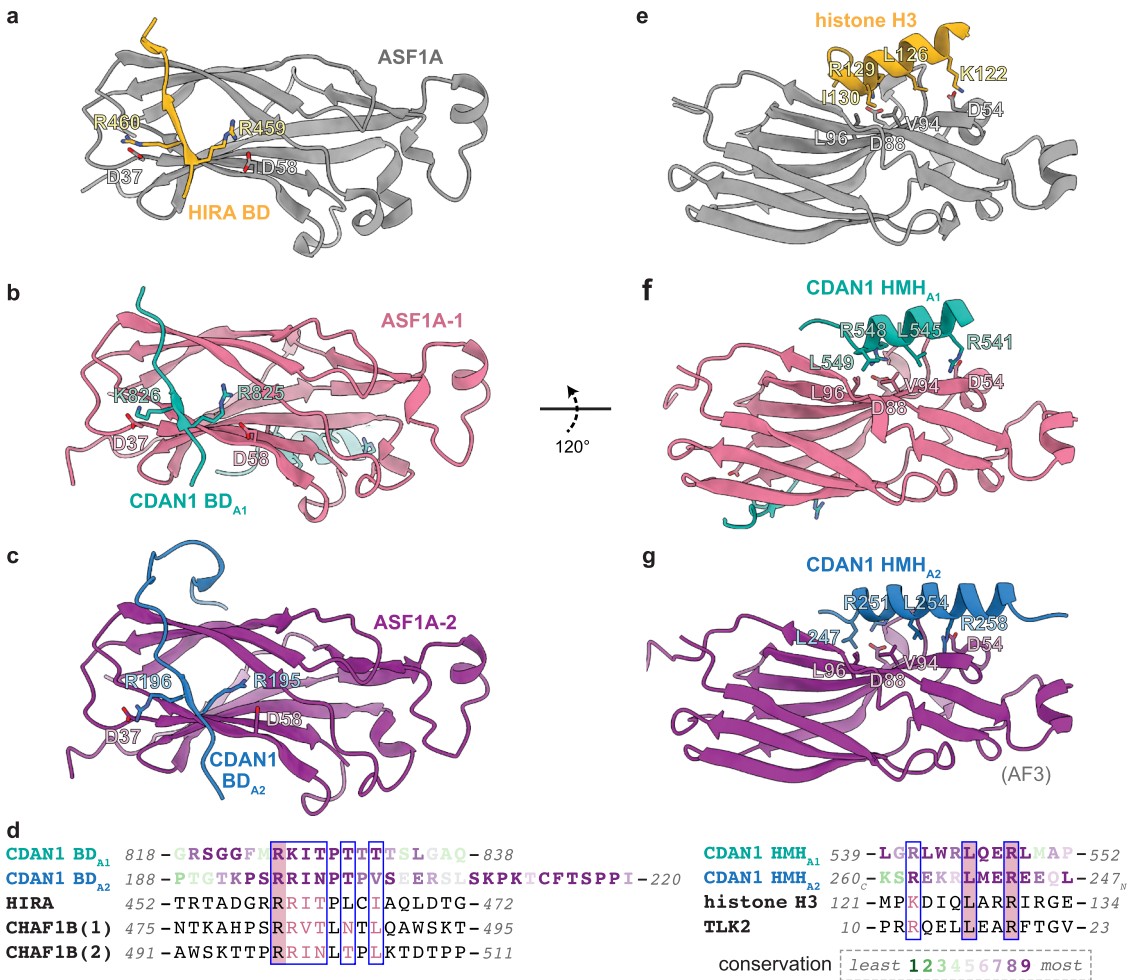

**Fig. 4 | CDAN1 interacts with ASF1 through multiple motifs. a** Structural model of the B-domain (BD) of human HIRA (gold; residues 458-466; PDB 2I32) bound to ASF1A (gray), with key BD interactions indicated. **b** Structural model of CDAN1 BD$_{A1}$ (teal) bound to ASF1A-1 (pink). **c** Structural model of CDAN1 BD$_{A2}$ (blue) bound to ASF1A-2 (purple). **d** Sequence alignments of CDAN1 B-domains (left), including the conserved C-terminal extension following BD$_{A2}$, and H3 mimic helices (right) with the indicated ASF1 interactors. CDAN1 sequences are colored according to

Consurf[77] conservation values as indicated. Conserved regions are boxed; identical residues are highlighted in pink; similar residues in aligned sequences are shown in pink text. **e** Structural model of the *X. laevis* histone H3 helix (gold; residues 121-136, PDB 2IO5) bound to human ASF1A (gray). **f** Structural model of CDAN1 HMH$_{A1}$ (teal) bound to ASF1A-1 (pink; 120° rotation relative to **b**). **g** Alphafold3 (AF3) prediction of CDAN1 HMH$_{A2}$ (blue) bound to ASF1A (purple).

Fig. 8a). Importantly, in all cases, CDIN1 association with the CDAN1 variants was not impacted, indicating that these mutations do not generally disrupt CDAN1 structure and that the effect of mutating each element is specific to ASF1 engagement. These results suggest that all ASF1 binding elements in CDAN1 described above contribute to ASF1 recruitment, albeit to different extents.

## ASF1A and ASF1B have different CDAN1 binding requirements

HMH$_{A1}$ appears to be required for ASF1B binding, as all CDAN1 variants with mutations in HMH$_{A1}$ failed to bind ASF1B (Fig. 5a, lanes 2,7,8 and 10, and Supplementary Fig. 8a). In contrast, interactions with other regions of CDAN1, such as BD$_{A2}$ and HMH$_{A2}$, appear sufficient for ASF1A to bypass mutations in HMH$_{A1}$ for CDAN1 engagement. ASF1A and ASF1B are over 70% identical (Supplementary Fig. 8b). The two paralogs diverge most at their N-terminal domains (NTD, residues 1-27), which contain eight different amino acids, and at their unstructured C-terminal tails, which are not resolved in our cryo-EM structures (Supplementary Fig. 8b,c). The histone chaperone HIRA has also been reported to bind ASF1A with higher affinity, in part by distinguishing the variable NTD of ASF1A and ASF1B using contacts outside of the canonical B-domain interaction[28]. To determine if the same region of ASF1 contributes to different CDAN1 binding requirements, we

generated NTD swaps of each ASF1 paralog and analyzed their association with ST-CDAN1 variants (Fig. 5b and Supplementary Fig. 8d).

Swapping the NTD of ASF1A to that of ASF1B diminished binding to WT CDAN1 (Fig. 5b and Supplementary Fig. 8d, top, compare lanes 1 and 2) and abrogated binding to CDAN1 containing mutations in HMH$_{A1}$ (Fig. 5b, top, lanes 3-6, and Supplementary Fig. 8d, top, lanes 3 and 4). Reciprocally, replacing the NTD of ASF1B with that of ASF1A enhanced binding to WT, HMH$_{A1}$*, and HMH$_{A1}$* BD$_{A1}$* CDAN1 (Fig. 5b, bottom, lanes 1-6, and Supplementary Fig. 8d, bottom, lanes 1-4). In comparison, mutating both HMH$_{A1}$ and BD$_{A2}$ disrupted binding to all ASF1 variants (Supplementary Fig. 8d, lanes 7-8), while all ASF1 variants could still interact with BD$_{A2}$* (Supplementary Fig. 8d, lanes 5-6) and HMH$_{A2}$* BD$_{A2}$* (Fig. 5b, lanes 7-8) CDAN1. These results indicate that if the HMH$_{A1}$ interface is disrupted, a compensatory interaction can occur between CDAN1 and ASF1A but not ASF1B in an NTD-dependent manner.

Inspecting the positions of the variable NTD residues in ASF1A in our structure showed that several reside near the B-domains and the conserved C-terminal segment extending from BD$_{A2}$ (Supplementary Fig. 8c and 9a). The NTD of both the distal and proximal ASF1A face the stacking interface between the two ASF1 molecules. In this configuration, the loop between the two β-strands of the NTD is near the

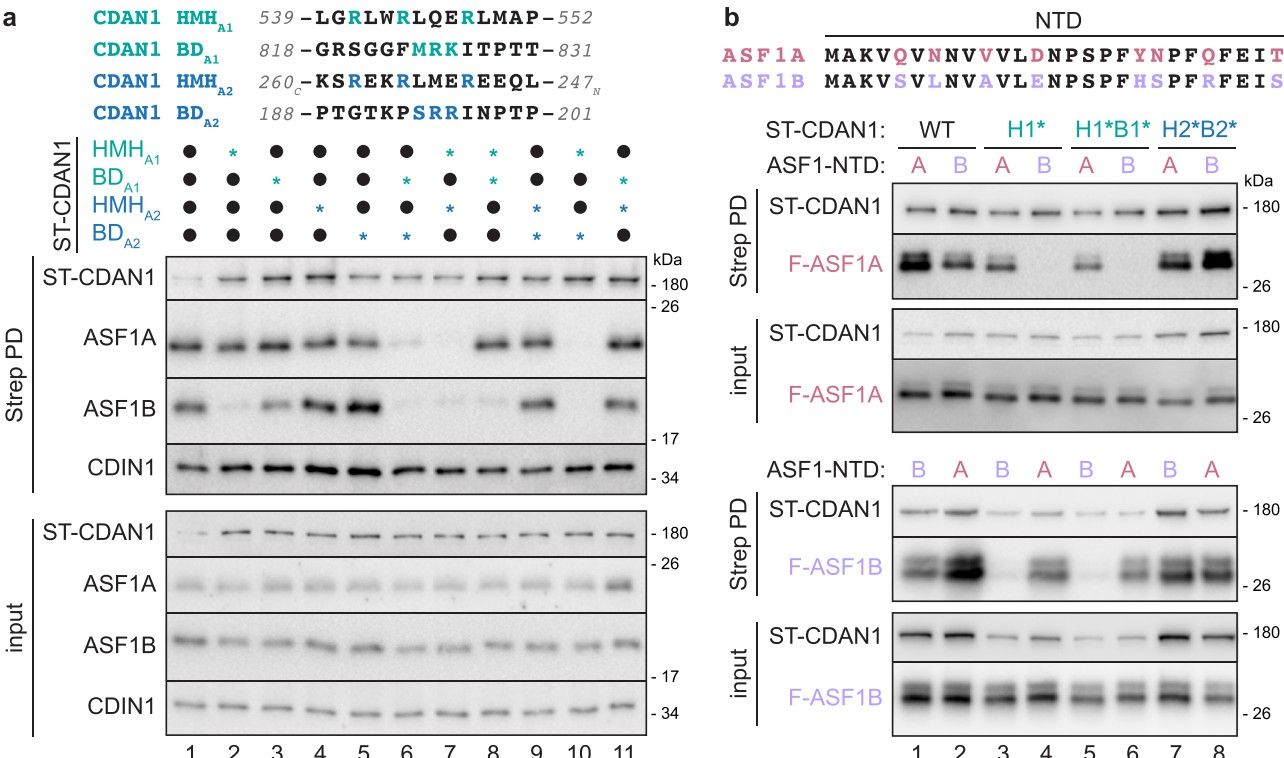

**Fig. 5 | ASF1A and ASF1B have different CDAN1 binding requirements.**
**a** Sequences of CDAN1 binding motifs (top); residues colored in teal or blue are mutated to alanines in the variants where indicated (*). ASF1B interaction with CDAN1 is more reliant on HMH$_{A1}$ than ASF1A (bottom). Flp-In 293 T-REx cells transiently transfected with Strep-tagged CDAN1 (ST-CDAN1) variants without (filled circles) or with (*) mutations in the indicated CDAN1 domains were lysed (input) and subjected to Strep-Tactin pulldowns (PD) before analysis by SDS-PAGE and immunoblotting; representative of 3 independent replicates. **b** The N-terminal

domain (NTD; first 27 amino acids) sequences of ASF1A and ASF1B, with divergent residues colored in pink for ASF1A or purple for ASF1B (top). Wildtype (WT) ST-CDAN1 or ST-CDAN1 variants containing mutations in HMH$_{A1}$ (H1*), BD$_{A1}$ (B1*), HMH$_{A2}$ (H2*), and/or BD$_{A2}$ (B2*) as in (**a**) were co-transfected with FLAG-tagged ASF1A (F-ASF1A; middle) or ASF1B (F-ASF1B; bottom) variants with the indicated NTD. Cell lysates before (input) or after Strep PD were analyzed by SDS-PAGE and immunoblotting; representative of 3 independent replicates. Source Data are provided as a Source Data file.

B-domain that interacts canonically with the other ASF1A (Supplementary Fig. 9a). Notably, Q23 of the distal ASF1A-2 (R23 in ASF1B) is ~4 Å from R825 of CDAN1 BD$_{A1}$, and Q23 of the proximal ASF1A-1 is ~6.5 Å from R195 of CDAN1 BD$_{A2}$. In addition, Q5 and N7 of the distal ASF1A-2 (S5 and L7 in ASF1B) are near the conserved extension following BD$_{A2}$ (Supplementary Fig. 8c and 9a). However, R23Q ASF1B was not sufficient to recoup binding to HMH$_{A1}$* ST-CDAN1 (Supplementary Fig. 9b). A double mutant, S5Q L7N, of ASF1B partially recovered binding to HMH$_{A1}$* ST-CDAN1 but not to HMH$_{A1}$* BD$_{A1}$* ST-CDAN1 (Supplementary Fig. 9c), which could interact with ASF1B containing a full ASF1A NTD swap (Fig. 5b, bottom, lane 6). This partial effect suggests that multiple interactions contribute to the different CDAN1 binding properties conferred by the ASF1A and ASF1B NTDs.

Considered together, our findings support a model in which HMH$_{A1}$, which packs against the structured MIF4G2 domain, forms a major docking site on CDAN1 for free ASF1 that is not bound to H3-H4. Binding to HMH$_{A1}$ would block the H3 binding site of ASF1, which may be further secured on CDAN1 by BD$_{A1}$. In addition, the long CDAN1 loop emerging from the MIF4G1 domain that contains both BD$_{A2}$ and the putative HMH$_{A2}$ can capture another ASF1 via interactions that likely favor ASF1A over ASF1B, ultimately establishing the stacked ASF1 conformation observed in our structures.

## Discussion

Our findings reveal the molecular basis for how CDAN1 can sequester multiple ASF1 molecules. In addition to the known B-domain (i.e., BD$_{A2}$) on CDAN1[21], we have identified a previously unappreciated second B-domain (BD$_{A1}$) on CDAN1 and two H3 mimic helices that

together permit one copy of CDAN1 to bind and occupy the histone chaperoning interface of two ASF1A molecules. The assignments of these ASF1-binding elements on CDAN1 are consistent with another recent structural study[52].

Although a CDAN1 dimer has four ASF1 binding sites, we do not observe the binding of a fourth ASF1 molecule in our recombinant CDAN1 complexes. Both SEC-MALS and our single-particle cryo-EM analysis consistently and orthogonally support a CDAN1:ASF1A ratio of 2:3 for independently purified complexes. This homogeneity may be specific to our recombinant complexes, and the composition and stoichiometry of endogenous complexes may be more variable. Because ASF1A concentrations do not appear to be limiting in our purifications, we speculate that conformational restraints upon the binding of a third ASF1 molecule may limit the engagement and stable stacking of a distal ASF1 on both arms of the CDAN1 dimer. Alternatively, regions of CDAN1 that are not resolved in our cryo-EM maps, such as the three-helix bundle or MA3 domain, may sterically hinder the recruitment of a fourth copy of ASF1. Consistent with these possibilities, we see evidence for both additional low-resolution densities and conformational heterogeneity in 3D variability analyses of our cryo-EM data (Supplementary Movie 1).

Future insights into CDAN1 complexes will require additional studies of CDIN1 and other domains of CDAN1. Notably, structural homology searches revealed that CDAN1 exhibits extensive homology to CNOT1 (Supplementary Fig. 5), the scaffolding component of the CCR4-NOT deadenylation complex[36,45–49]. While human CNOT1 (2376 residues) is larger than CDAN1 (1227 residues), an internal ~1000 residue segment of CNOT1 folds into a series of four domains that are

structurally homologous or similar to the entirety of CDAN1. These four domains are also encoded in the same order in both proteins. Structural homology searches of the two MIF4G domains and the DUF3819 three-helix bundle of CDAN1 in isolation against the predicted human proteome consistently identify the corresponding domain in CNOT1 as a top match[45]. The fourth domain is a member of the HEAT family that forms an MIF4G-like domain in CNOT1 but more closely resembles an MA3 domain in CDAN1. MIF4G domains are commonly found in translational regulators, often in tandem with other shared domains, such as MA3[44,47,55]. However, no other MIF4G proteins contain predicted homology to DUF3819[45]. These findings suggest that CDAN1 and CNOT1 may have evolved from a common ancestor and have related functions. The possibility that CDAN1 controls substrate access for the putative nuclease function of CDIN1, as an analogy to the CNOT1-mediated coordination of the Ccr4/Caf1 nucleases during deadenylation, is an intriguing comparison for future investigation.

The simultaneous use of multiple B-domains and H3 mimic helices by CDAN1 to engage ASF1 is unique among human ASF1 interactors. ASF1 paralogs can interact with other chaperones, such as HIRA and the p60 subunit of CAF-1 (CHAF1B), via a B-domain[28,29,31,56]. However, unlike CDAN1, these factors engage ASF1 bound to an H3-H4 heterodimer[57], primarily in the nucleus, as part of productive nucleosome assembly pathways during replication-independent (HIRA) or replication-dependent (CAF-1) deposition[58,59]. Other factors have also been reported to bind ASF1 through the H3-binding interface. Similar to CDAN1, yeast Rad53, and Rtt109 use both a B-domain and an additional region to bind ASF1 and compete with histone binding[30,32]. However, these interactions appear to occur only in yeast[30,60]. TLK2 also uses an H3 mimic helix to engage ASF1 and mediate ASF1 phosphorylation, which is suggested to promote histone binding and nucleosome assembly[54]. In contrast, CDAN1 is reported to sequester ASF1 in the cytosol when overexpressed[21]. Considered together with the HMH interactions that occupy the H3-binding site of ASF1, CDAN1 is distinctive in its ability to simultaneously engage multiple ASF1 molecules and potentially divert them from promoting histone deposition.

Moreover, we found that although CDAN1 binds both ASF1A and ASF1B, there are different requirements for the interaction of each paralog. In comparison, CAF-1 binds both ASF1 paralogs[61], while HIRA specifically binds ASF1A[28]. CDAN1 may, therefore, impact the function and binding of other interactors to each ASF1 paralog to different extents. In most dividing cells, the expression of CDAN1 is low relative to other ASF1 interactors. For example, in HEK293T cells, CAF-1 is approximately equimolar with ASF1A or ASF1B, which are each ~10-fold more abundant than CDAN1[62]. Similarly, HIRA is several-fold more abundant than CDAN1. Thus, although CDAN1 can bind multiple ASF1 molecules, it is unlikely to engage a significant proportion of ASF1 in most cells, even though the essential nature of CDAN1 suggests it nonetheless has a critical function in most cell types.

However, if the balance of the levels of ASF1 and its interactors shifts, CDAN1 may assume an outsized role by quantitatively competing with histone deposition pathways. Such dynamics may arise during terminal erythropoiesis when maturing erythroblasts undergo extensive proteome remodeling and chromatin condensation in preparation for organellar clearance[63]. At the proerythroblast stage, the levels of ASF1 interactors are altered, with approximately equal levels of CAF-1 and CDAN1 that are both expressed ten-fold in excess of HIRA[64]. Under such conditions, CDAN1 may sequester a substantial proportion of ASF1 in an incompletely defined process that is significant enough to cause pathogenic defects in chromatin condensation when CDAN1 function or expression is disrupted through mutations. The molecular-level insights uncovered in our study will drive forward the understanding of these functional roles of CDAN1 by informing detailed dissections of CDAN1 complexes in different physiological contexts.

## Methods

### Plasmids and antibodies

The initial construct encoding HaloTag was a gift from the Adelman lab[65]. The CDAN1 cDNA was purchased from Horizon Discovery/Dharmacon (MHS6278-202833430); a single point mutation (S421F) was corrected by Gibson assembly. cDNA sequences for ASF1A, ASF1B, and CDIN1 were synthesized by IDT. cDNA sequences were cloned into a pCDNA3.1 mammalian expression vector encoding an N-terminal 3xFLAG, 3xHA, or 3xStrep-II-TEV tag, where indicated. Mutagenesis was performed with either Gibson assembly or Phusion mutagenesis to generate Strep-CDAN1 HMH$_{A1}$* (R541A/R544A/R548A), Strep-CDAN1 HMH$_{A2}$* (R251A/R255A/R258A), Strep-CDAN1 BD$_{A1}$* (M824A/R825A/K826A), or Strep-CDAN1 BD$_{A2}$* (S194A/R195A/R196A), combinatorial BD and HMH Strep-CDAN1 mutants as described in Fig. 5a and Supplementary Fig. 8a, FLAG-ASF1A N > B (8 variant residues in the 27 residue NTD swapped to their ASF1B counterparts: Q5S/N7L/V10A/D13E/Y19H/N20S/Q23R/T27S), FLAG-ASF1B N > A (8 variant residues in the 27 residue NTD swapped to their ASF1A sequence: S5Q/L7N/A10V/E13D/H19Y/S20N/R23Q/S27T), FLAG-ASF1B RQ (R23Q), FLAG-ASF1B QN (S5Q/L7N), and Strep-CDAN1$_C$ (residues 1008–1227).

For endogenous tagging, donor templates were constructed in a pKI plasmid (a gift from H. Chino) encoding 500 bp homology arms, amplified from gDNA for CDAN1 or CDIN1, flanking the HaloTag-FLAG insert followed by a stop codon. The PAM site was mutated in the template to prevent self-cleavage. Guide RNAs targeting CDAN1 (CTGGTGCAATGCCCAAGGCA) or CDIN1 (GCATAAGGTGA-CAATGTTCG) were designed using Benchling (https://benchling.com) and inserted into the pX459 plasmid[66] using restriction enzyme cloning.

Antibodies used for blotting: HRP-conjugated anti-FLAG M2 (Sigma, A8592, 1:10,000), HRP-conjugated Strep-Tactin (Bio-rad, 1610381, 1:5000), HRP-conjugated anti-HA (Cell Signaling Technology, 2999S, 1:5000), anti-CDIN1 (Abcam, ab215190, 1:1000), anti-CDAN1 (Bethyl, A304-952A, 1:1000), anti-histone H3 (Proteintech, 17168-1-AP, 1:5000), anti-histone H4 (Proteintech, 16047-1-AP, 1:1000), anti-DNAJC9 (Proteintech, 25444-1-AP, 1:1000), anti-ASF1A (Cell Signaling Technology, 2990S, 1:1000), and anti-ASF1B (Proteintech, 22258-1-AP, 1:1000). Antibodies used for immunofluorescence: CDIN1 (Atlas Antibodies, HPA061023, 1:50).

### Cell culture and cell line generation

Flp-In 293 T-REx cells were cultured in Dulbecco's Modified Eagle's Medium (DMEM) supplemented with 10% fetal bovine serum (FBS) at 37 °C and 5% CO$_2$. Expi293F cells were cultured in Expi293 medium (Gibco A1435101) at 37 °C and 8% CO$_2$ with shaking at 120 r.p.m.

To generate endogenously tagged cell lines, Flp-In 293 T-REx cells were reverse co-transfected with pKI and individual pX459 plasmids using TransIT-293 (Mirus) according to the manufacturer's instructions in media containing 690 nM Alt-R™ HDR Enhancer V2 (IDT, 10007910). After 48 hr, cells were placed under 2 µg/mL puromycin (Gibco, A11138) selection for an additional 48 hr. After selection, cells were labeled with 100 nM HaloTag TMR ligand (Promega, G8252) for 30 min at 37 °C to detect insertion of the donor template. Single clones with positive fluorescent signals were sorted into 96-well plates using a Sony SH800 Sorter. Cell lysates were isolated from each colony, and tagging was confirmed by immunoblotting to detect the anticipated size shift of the endogenous protein. The epitope recognized by the CDAN1 antibody is blocked by the HaloTag. All clones were also validated by PCR and Sanger sequencing of the endogenous locus.

For siRNA treatment, Flp-In 293 T-REx cells were reverse transfected with 10 nM siRNA (siNeg, siCDAN1, or siCDIN1 [Horizon Discovery/Dharmacon, D-001810-10-20, L-015478-00-0005, or L-014906-02-0005]) using Lipofectamine RNAiMAX (Invitrogen, 13778-150) according to the manufacturer's instructions. For Supplementary Fig. 1d, cells were plated in a 12-well glass bottom plate (Mattek,

NC0190134). After 48 hr, cells were either lysed for immunoblotting or washed with phosphate-buffered saline (PBS) and fixed for immunofluorescence. For targeted degradation, cells were treated with either 500 nM HaloProtac3 (HP3; a gift from the Adelman lab or purchased from Promega, GA3110) or DMSO/Ent-HP3 (Promega, GA4110) for the specified amount of time.

## Co-immunoprecipitations

In general, Flp-In T-REx 293 cells were co-transfected with 1 µg plasmid per construct in a 6-well plate using TransIT-293 according to the manufacturer's instructions. 16–24 hr after transfection, the cells were collected in cold PBS, lysed in lysis buffer (50 mM HEPES pH 7.5, 100 mM KOAc, 2.5 Mg(OAc)$_2$, 1% digitonin with 1 mM dithiothreitol [DTT], 1 × cOmplete protease inhibitor cocktail [PIC; Roche, 1187358] with or without 1 × PhosStop (Roche)) for 10 min on ice, and clarified by centrifugation at 21,130 × g for 10 min at 4 °C. Lysis buffer for Strep-Tactin pulldowns (PD) also contained BioLock (IBA, 2–0205). Normalized lysates were added to immunoprecipitation (IP) buffer (50 mM HEPES pH 7.5, 100 mM KOAc, 2.5 Mg(OAc)$_2$, 1% Triton X-100) containing either anti-FLAG M2 agarose (Sigma, A2220) or Strep-Tactin Sepharose HP resin (Cytiva, 28-9355-99) and incubated while rotating for 1 hr at 4 °C. After rotation, samples were washed 3 times with IP buffer and eluted in protein sample buffer.

To immunoprecipitate endogenously tagged CDAN1 or CDIN1, 12 15 cm dishes of each cell line were lysed in 12 mL IP buffer supplemented with 1 mM DTT, 1 × PIC, and 1 × PhosStop and incubated with anti-FLAG M2 agarose resin. Beads were washed, eluted in protein sample buffer, and analyzed using SDS-PAGE and Coomassie staining.

**Cellular fractionations.** An approximately equal number of parental or endogenously tagged CDIN1-HF or CDAN1-HF Flp-In 293 T-REx cells were lysed using the NE-PER Nuclear and Cytoplasmic Extraction Kit (Thermo Scientific) supplemented with 1 × PIC and 1 × PhosStop. The cytosolic fraction was extracted following the manufacturer's instructions. The remaining pellet containing the nuclei was washed and lysed in protein sample buffer with Benzonase (Millipore).

## Live-cell imaging and immunofluorescence

For live-cell imaging, cells were plated in either an 8-well chambered cover glass (Cellvis C8-1.5H-N) or a 12-well glass bottom plate. After 48 hr, cells were treated with 100 nM JFX650 HaloTag Ligand (a gift from A. Mizrak, the Harper lab, and the Lavis lab) for 1 hr at 37 °C. Immediately before imaging, cells were stained with Hoechst (Invitrogen, H3570), washed, and placed in FluoroBrite DMEM (Gibco) supplemented with 10% FBS. Cells were mounted either in an OkoLab cage or stage-top microscope incubator heated to 37 °C with 5% CO$_2$ for imaging.

For fixed imaging, cells grown on glass bottom plates were washed in PBS and fixed with 4% PFA in PBS for 10 min. Cells were then washed 3 times, permeabilized with 0.1% Triton for 5 min, then incubated with blocking buffer (10% FBS or 3% BSA in PBS with 0.05% Tween 20 [PBSt]) for 1 hr at room temperature. The primary antibody was diluted in blocking buffer and added at room temperature for 1 hr or 4 °C overnight. After three PBSt washes, cells were incubated with Alexa Fluor 564-conjugated goat anti-rabbit IgG secondary antibody (1:500; Jackson ImmunoResearch, 111-585-003) in blocking buffer for 1 hr before labeling with Hoechst, then washed in PBSt.

All images were collected with a confocal Yokagawa CSU-X1 spinning disk confocal on a Nikon Ti inverted microscope equipped with either a Nikon Plan Fluor 40x Oil DIC H N2 (Supplementary Fig. 1d), Plan Apo 60 × Oil DIC H objective (Fig. 1d), or Plan Apo λ 100x Oil objective (Supplementary Fig. 1e, f). Fluorescence was excited with solid-state lasers at 405 nm (80 mW), 561 nm (65 mW), or 640 nm (60 mW) and collected using ET455/50 m, ET620/60 m, or ET700/75 m emission filters (Chroma), respectively. Images were acquired with a Hamamatsu ORCA-Fusion BT sCMOS camera controlled with NIS elements 5.21 software. Brightness and contrast were adjusted identically for compared image sets in Fiji[67].

## Purification of CDAN1 complexes

ST-CDAN1 complexes were purified via co-expression with F-ASF1A (C:A) or both F-CDIN1 and HA-ASF1A (C:C:A), or via co-expression of truncated ST-CDAN1$_C$ with F-CDIN1 (C:C), from Expi293 cells. Cells were seeded at 2.5 million cells/mL. 24 hr later, cells were diluted to 3 million cells/mL, and 0.5–1 µg/mL of each construct was transfected with 5 µg/mL polyethyleneimine (PEI-25K; Polysciences, 23966). 24 hr after transfection, 3 mM sodium valproate and 0.45% glucose were added to the cells. After 48 hr, the cells were collected, washed once in cold PBS, and either flash frozen and stored at − 80 °C for later processing or immediately lysed in 1 mL IP buffer supplemented with 1 mM DTT and 1 × PIC per 10 mL culture. Lysates were clarified by centrifugation at 15,000-21,130 × g for 10–20 min at 4 °C. The supernatant was incubated with anti-FLAG resin for one hr at 4 °C while rotating. The lysate was then passed through a column by gravity flow, washed with IP buffer, then with wash buffer (WB; IP buffer with 1 mM DTT but lacking Triton-X 100), and eluted using 3 × FLAG peptide (APExBIO) in WB. The eluate was then incubated with Strep resin in WB for one hr at 4 °C while rotating. After incubation, the sample was eluted using 10 mM desthiobiotin in WB.

## SEC-MALS

C:A and C:C:A complexes were purified as described. The elutions were pooled and applied to a Superose 6 10/300 GL column (Cytiva) pre-equilibrated in 1 × PBS, pH 7.4 (Corning, 46-013-CM) with 1 mM DTT, and 0.2 mL fractions were collected. Peak fractions were pooled, concentrated, and 200 µg of each complex at 5 µM were applied to a pre-equilibrated SRT SEC-300 column (Sepax) connected to a DAWN HELEOS II Multi-Angle Light Scattering detector (Wyatt Technology) followed by an Optilab T-rEX Refractive Index Detector (Wyatt Technology) and a WyattQELS detector for Dynamic Light Scattering (Wyatt Technology). Data was analyzed in Astra 7 software (Wyatt Technology). Protein concentration was measured using the refractive index; a refractive index increment (dn/dc) of 0.185 was used in MALS calculations. BSA was used as a standard (Thermo Fisher, 23209).

## Cryo-EM sample preparation and data collection

C:C:A complexes purified as described above were cross-linked with 250 µM BS3 (Thermo Fisher Scientific, 21580) for 15 min on ice and quenched with 2.5 mM Tris pH 7.6. 3 µL of the crosslinked sample at 1 mg/mL was applied to glow-discharged 0.6/1 UltrAuFoil 300 mesh grids (Quantifoil) and frozen in liquid ethane using a Vitrobot Mark IV (Thermo Fisher Scientific) set at 4 °C and 100% humidity with a blot time of 3 s, a blot force of + 8, and a wait time of 30 s. CDAN1 C-term:CDIN1 (C:C) complexes were prepared as described above except crosslinked and plunge frozen at a final concentration of 0.78 mg/mL.

Datasets were collected using a Titan Krios (Thermo Fisher Scientific) operating at 300 kV and equipped with a BioQuantum imaging filter with a 20-eV slit width and a K3 direct electron detector (Gatan) in counting mode at a nominal magnification of × 105,000 corresponding to a calibrated pixel size of 0.825 Å. Semi-automated data collection was performed with SerialEM. For the C:C:A dataset, 2.708-second exposures were fractionated into 49 frames, resulting in a total exposure of 50.3 electrons/Å$^2$. The defocus targets were − 1.3 to − 2.3 µm. For the C:C dataset, 2.597-second exposures were fractionated into 50 frames, resulting in a total exposure of 52.72 electrons/Å$^2$. The defocus targets were − 1.2 to − 2.2 µm.

## Image processing and model building

Data processing was performed using cryoSPARC[68] v4.3.1. Patch-based motion correction and CTF estimation were applied during cryoSPARC

Live, and micrographs with severe contamination were removed. For the C:C:A complex, 3250 micrographs were subjected to automated particle picking using templates generated from blob-based picking. Particles were initially extracted with a box size of 300 and downsampled to a box size of 128 for 2D classifications to remove junk particles. After ab initio reconstructions and heterogeneous refinements, 256,296 particles were unbinned and subjected to CTF refinement, local motion correction, and non-uniform and local refinements with C1 symmetry to produce a reconstruction of a CDAN1 dimer with two stacked ASF1A molecules at an overall resolution of 3.0 Å. The same particle set was also subjected to a non-uniform refinement with C2 symmetry that was used for symmetry expansion. Two rounds of 3D classification were performed with the expanded particle set with C1 symmetry using masks focused on the stacked ASF1A molecules associated with one CDAN1. This clearly demonstrated the occupancy of one or two ASF1A molecules. Finally, 122,061 particles displaying occupancy for two ASF1A molecules were subjected to a final masked local refinement.

To better visualize a CDAN1 dimer associated with 3 ASF1A molecules, a separate processing pipeline was employed using particles extracted in a larger box size of 500 and downsampled to a box size of 100 for initial ab initio refinement and 3D classification resulting in 25,056 particles with clear density for 3 ASF1A molecules that were unbinned and subjected to a final round of non-uniform refinement to produce a reconstruction at an overall resolution of 3.5 Å. 3D Variability Analysis (3DVA) was performed in cryoSPARC on the input particles used for 3D classification for two components with a filter resolution of 6 Å; the first component is shown in Supplementary Movie 1.

Half maps of C:C:A complex reconstructions were post-processed using DeepEMhancer[69] for interpretation. Alphafold2[70] models of human CDAN1 (Q8IWY9) and ASF1A (Q9Y294) were used as initial models that were fitted as rigid bodies into the cryo-EM map of the C:C:A complex in ChimeraX[71] v1.6. CDAN1 BD$_{A1}$ and BD$_{A2}$ placements were modeled using regions isolated from an Alphafold3[51] prediction of one CDAN1 with two ASF1A molecules, aligned to the ASF1A models that were rigid body fitted into the map. The model was then manually adjusted in Coot[72] v0.9 with multiple rounds of Phenix[73] real space refine with manual adjustments in Coot in between each round. Refinements were first performed using a model with one copy of CDAN1 and two copies of ASF1A against the map focused on the two stacked ASF1A molecules, and then with the whole complex containing a CDAN1 dimer and three ASF1A molecules. Data processing was supported by software packages installed and configured by SBGrid[74]. Figure panels were made with ChimeraX. Initial multiple sequence alignments were generated using Clustal Omega[75] and visualized using ESPript 3.0[76]. CDAN1 sequence conservation was analyzed using the ConSurf[77] server.

For the C:C (CDAN1 C-Terminal:CDIN1) complex, 2271 micrographs were subjected to automated particle picking using templates initially generated from blob-based picking and extracted with a box size of 400 and downsampled to 200 for 2D classification. Next, template-based particle picking was performed using selected 2D classes (templates representing 170,087 particles). Particles extracted with a box size of 256 and downsampled to 128 were subjected to several rounds of 2D classification, including a second round of template-based picking. 421,269 particles were used for ab-initio reconstructions into 10 classes, followed by heterogenous refinement for each class. The class with the largest percentage of particles was used as a reference volume for non-uniform refinement with 355,895 particles obtained from a rebalance 2D job performed on the input particle stack, which was used to balance overrepresented particle views in the dataset using the following parameters: 28 initial templates, 20 superclasses, and a rebalance factor of 0.5. After additional 2D classification, 195,239 particles from an additional rebalance 2D job

were subjected to non-uniform refinement. The Alphafold3[51] prediction of the C:C complex was fitted as a rigid into the density in ChimeraX for visualization.

## Reporting summary
Further information on research design is available in the Nature Portfolio Reporting Summary linked to this article.

## Data availability
The EM maps and models generated in this study have been deposited in the EMDB under accession code EMD-45959 (CDAN1 dimer with three ASF1A), EMD-45960 (CDAN1 dimer with two ASF1A), EMD-45961 (CDAN1 with two ASF1A), and the PDB under accession code 9CVC. The EM map of the CDAN1$_C$:CDIN1 complex was not used for modeling and is available in Figshare [https://doi.org/10.6084/m9.figshare.26414005]. All other data are available within the article and its Supplementary Information. Source data are provided in this paper.

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

## Acknowledgements

Cryo-EM screening and data collection were performed at the Harvard Center for Cryo-Electron Microscopy (HC2EM). Data processing was supported by SBGrid. Light microscopy was performed at the Core for Imaging Technology and Education (CITE) at Harvard Medical School (HMS). SEC-MALS was performed at the Center for Macromolecular Interactions (CMI) at HMS. We thank H. Chino for guidance with endogenous tagging and help with cell sorting, M. Yip for guidance with cryo-EM sample preparation and processing, M. McKenna and D. Sherpa for critical reading, and Shao lab members for useful discussions. This work was supported by NIH F31HL157976 (S.F.S.), NIH DP2GM137415 (S.S.), and a Packard Fellowship (S.S.). SS is an Investigator with the Howard Hughes Medical Institute.

## Author contributions

S.F.S. conceived the project and performed all investigations. S.S. supervised the project and acquired funding. S.F.S. and S.S. wrote the paper.

## Competing interests

The authors declare no competing interests.
