## [Transparent Peer Review file · Nature Communications]

Mechanism of ASF1 Engagement by CDAN1

Corresponding Author: Dr Sichen Shao

Version 0:

Reviewer comments:

Reviewer #1

(Remarks to the Author)

Sendor and Shao have presented a cryo-EM structure of the CDAN1-Asf1 complex, highlighting its role in sequestering Asf1 from histone binding. As Asf1 is a highly conserved histone chaperone with key functions, this structure holds significant interest especially in the chromatin research field. The observed structural resemblance to Ccr4-Not adds another intriguing layer of complexity.

The structure determination of the CDAN1-Asf1 complex appears robust, further supported by biochemical analyses. While I support the publication, I believe there are several areas where the authors can enhance the presentation of their findings: 1) given another CDAN1-Asf1 structure in bioRxiv as authors' referenced revealed a symmetric structure, I wonder whether SEC-MALS can definitively conclude the 2:3 stoichiometry for the C:A complex and exclude the possibility of the other stoichiometries (e.g., 2:2 or 2:4)? I also wonder why the 4th copy of Asf1 would not bind to the second CDAN1? 2) Fig. 3 was very helpful to understand authors' structure, however, other figures (Fig. 4 Fig. S7d, Fig. S8b etc.) are likely viewed from different orientations. Clearer connections between these figures would greatly enhance their utility in conveying the structural relationships. 3) Figure S7 contains critical information but may overwhelm general readers. Especially it would be helpful to clarify which regions are included in the final deposited structure. 4) The authors have not deposited the CDAN1-CDIN1 structure in EMDB or PDB, making it difficult for readers to assess or even view the structure. Furthermore, the manuscript lacks details on the statistics and refinement procedures. While 6 Å resolution should allow for visualization of α -helices, Figure S6 does not offer sufficient clarity for such assessments.

Other minor comments are as follows:

P5: This domain architecture, particularly the distinctive presence of the three-helix bundle, resembles the central portion of the CNOT1 protein that serves as a scaffold in the CCR4-NOT deadenylation complex^{36,45–48} (Extended Data Fig. 5c-f). This observation is intriguing, but it would benefit from being included in the Discussion section, where it could be explored in greater depth.

P6: BDA2 also extends into additional conserved residues that wrap around ASF1A-2, which may help stabilize the position of ASF1A-2 stacked on top of ASF1A-1.

BDA2 is apparently much longer than the canonical B-domain. Why not show key interactions between the extended C-terminal region and Asf1 in Fig. 4B? Also, as far as I know, the canonical B-domain is always stabilized by forming a beta-sheet. Why not refine the structure by applying some constraints to enforce the hydrogen-bonding between beta-strands. It is much easier for readers to find the B-domain if we see its corresponding b-strand arrow.

P7: Specifically, a loop (residues 469-476) associated with CDAN1 MIF4G2 would clash with an H4 tail positioned on ASF1A-1 and the extended BDA2 loop would clash with an H4 tail positioned on ASF1A-2 (Extended Data Fig. 7d).

It would benefit by relating this to the entire structure. Also, for the left panel of Fig. S7d, some constraints to enforce b-strands of Asf1 should be applied during refinement.

P8: The histone chaperone HIRA has also been reported to bind ASF1A with higher affinity, in part by distinguishing the variable NTD of ASF1A and ASF1B that lie proximal to the Bdomain interface²⁸ (Extended Data Fig. 8b).

The reference 28 has shown that HIRA does not distinguish between ASF1a and ASF1b through the interaction between the HIRA B-domain and the ASF1 core domain.

This biochemical analysis is intriguing, but it would be more informative if the authors clearly show which residues of Asf1a, but not Asf1b, are critical for binding. The side chains are difficult to discern in Figure S8. Additionally, it's unclear why BDA1

and BDA2 are positioned differently between the left and right panels. Readers would benefit from clearer guidance on these points.

Reviewer #2

(Remarks to the Author)

The authors examine the potential mechanism by which Asf1 histone chaperone function might be inhibited by Codanin-1 (CDAN1) using biochemical and structural approaches. They first provide solid evidence that CDAN1, CDIN1 and ASF1 can be found in a complex using immunoprecipitation experiments from cells expressing endogenous levels of the tagged proteins. (Figure 1). The complexes are found in the cytosol, not the nucleus and despite containing both ASF1A and ASF1B, do not contain histone H4. Furthermore, the stability of CDIN1 depends on the presence of CDAN1. In Figure 2 the composition of the complex is assessed using overexpressed proteins, immunoprecipitation and size-exclusion chromatography. Although a CDAN1-CDIN1-ASF1 complex can be purified, the CDAN1-ASF1 complex (lacking CDIN1) was also stable. Notably histone H3 was excluded from both of these complexes. The SEC suggests that the stoichiometry of the two complexes is 2 CDIN1 : 2 CDAN2 : 3 ASF1 and 2 CDAN1 : 3 ASF1, respectively. Next, the cryo-EM structures were determined for the CDIN1-CDAN1-ASF1A complex (Figure 3). However, CDIN1 and part of CDAN1 were not visible, but overall the structure supports the observed stoichiometry (Fig. 2). The structure reveals that the dimer of CDAN1 binds to 2 or 3 ASF1A. The interactions (Figure 4) appear to be between ASF1A and B-domain segments (BDA1 and their newly identified BDA2) as well as histone helix 3 mimic domains (HMHA1 and HMHA2, deduced from AlphaFold and the EM models). These observations raise the possibility that CDAN1 might be able to block ASF1-histone interactions through competition via H3-interface and the H4 binding site. The specificity of CDAN1 for ASF1A versus ASF1B was examined using alanine amino acid substitutions at key sites within in BDA2 and HMHA1 as well as ASF1A-ASF1B specificity swap substitutions in the N-terminal region (Figure 5). These results suggest that there is selectivity of CDAN1 motifs for ASF1B, but not ASF1A. These results also provide some validation of the Cryo-EM models.

This study presents interesting and novel findings about the mechanism by which CDAN1 could inhibit ASF1 histone chaperone activity. The structure of the CDAN1 complex with CDIN1 and ASF1 supports a 1:1:>1 stoichiometry and reveals novel ASF1 binding regions in CDAN1, which would be expected to be competitive with histone H4 and H4. The results provide generally adequate support for the authors conclusions, except for the title. Specifically, the manuscript could be strengthened with more complete mutagenesis studies and competition experiments.

1) A major conclusion is that CDAN1 can bind to 2 ASF1 molecules using 2 B domain (BDA2) and H3- helix 3 mimics (HMHA1 and HMHA2). To this reviewer, the cryo-EM is not completely clear about these interactions as the density seems to be weak and the placement of the binding domains is largely through modeling (Extended data figures). Given this possibility for ambiguity in the placement of the binding motifs, further structure validation is needed. The mutagenesis studies only addressed BDA2 and HMHA1 and would benefit from also testing combinations of mutants of BDA1 and HMHA2 and the use of disruptive mutations in addition to alanine.

2) The possible mechanism of inhibition of ASF1 function is interesting, but there isn't strong direct evidence that CDAN1 actually inhibits histone binding to ASF1. The binding affinity of ASF1 for histones is very strong and this study does not test whether CDAN1 can compete with histones for ASF1. Further studies including binding affinities and competition experiments would be needed to prove this statement, which is the title of the manuscript.

3) The observation that the BD and HMH interactions with the multiple ASF1 proteins is not symmetric and that the stoichiometry is not also symmetric (1:1:2 or 2:2:4) is very surprising. SEC was the only method used to support the unusual stoichiometry (2:2:3) and this has error associated with it. Please explain how any of the EM approaches (grid preparation – complex falling apart; or averaging, etc.) might have influenced the appearance of the complex. Is there a better explanation for this unusual stoichiometry? Equilibrium analytical ultracentrifugation would provide a more accurate estimate of the size of the complex.

4) There are results derived from only two (not 3 or more) independent replicates.

5) Figure 2: what are the doublets in Fig. 2a and b for F-CDIN1, HA-ASF1A and F-ASF1A?

Version 1:

Reviewer comments:

Reviewer #1

(Remarks to the Author)

The authors have fully addressed the points and concerns I raised. I believe the structure is now much better presented, benefiting all readers.

Reviewer #2

(Remarks to the Author)

The authors have adequately addressed my concerns. Thank you!

Reply to Reviewers

We thank the reviewers for their constructive feedback and suggestions, which have improved our manuscript. We incorporated revisions based on their comments and have highlighted the main changes below. In addition, we include point-by-point replies to their comments.

Main changes:

- Additional mutagenesis analyses showing that all four ASF1 binding elements on CDAN1 (BD_{A1} , BD_{A2} , HMH_{A1} , and HMH_{A2}) contribute to ASF1 binding (new **Fig. 5** and **Extended Data Fig. 8a,d**)
- Additions and adjustments to figure panels with updated structural models (e.g., **Fig. 3d, 4, and Extended Data Fig. 4d, 7, 8c, 9a**) to show the β -strand of the B-domains and associated interactions, as well as clearer relations between the overall structure and specific views.
- Modifications to the title and text to tone down the implication that CDAN1 directly inhibits ASF1 chaperoning activity by competing with H3-H4 binding.

Reviewer #1

1) given another CDAN1-Asf1 structure in bioRxiv as authors' referenced revealed a symmetric structure, I wonder whether SEC-MALS can definitively conclude the 2:3 stoichiometry for the C:A complex and exclude the possibility of the other stoichiometries (e.g., 2:2 or 2:4)?

We note that the CDAN1-ASF1 cryo-EM structure reported in the current version of the Jeong, Frater, et al. preprint (doi: 10.1101/2024.07.10.602876) **applied C2 symmetry** to generate their cryo-EM map. This is commonly used to enhance the resolution of one symmetric unit by effectively doubling the number of 'particles' contributing to the reconstruction. However, it **assumes that the molecule is symmetric and enforces symmetry on the reconstruction**. We also applied C2 symmetry as an intermediate step in our processing (**Extended Data Fig. 3**, rightmost pipeline). This intermediate reconstruction showed strong density for the proximal ASF1A-1 and weaker density for the distal ASF1A-2 because both sides are an identical average of each CDAN1 monomer complex unit; it is also similar to the reconstruction presented in Jeong, Frater, et al. Because of this clear heterogeneity, we **only used the C2 reconstruction to perform symmetry expansion** on the particles (see <https://guide.cryosparc.com/processing-data/all-job-types-in-cryosparc/utilities/job-symmetry-expansion>) so that we can focus specifically on every CDAN1 monomer unit for further classification **to resolve symmetry-breaking features**. **Focused classification of the expanded particles showed a ~50:50 split** between CDAN1 monomers bound to two versus one ASF1, consistent with an asymmetric complex. This also matches our observations in unbiased refinements without any symmetry applied (**Extended Data Fig. 3**, leftmost pipeline) and the SEC-MALS data.

In the new **Extended Data Fig. 2c** and below, we show the molar mass ranges for CDAN1:ASF1A complexes purified without (C:A) or with (C:C:A) CDIN1 determined by SEC-MALS and the theoretical molar mass (including tags) for ratios of 2:2, 2:3, or 2:4 CDAN1:ASF1A or 2:2:2, 2:2:3, or 2:2:4 CDAN1:CDIN1:ASF1A.

Sample	SEC-MALS molar mass	Range (\pm uncertainty)	Theoretical Molar Mass C:(C:)A stoichiometry			Error (2:(2:3))	Polydispersity
			2:(2:)2	2:(2:)3	2:(2:)4		
C:A	364.3 \pm 0.40%	362.8 - 365.8	331.7	357.7	383.7	1.8%	1 \pm 0.50%
C:C:A	432.1 \pm 0.40%	430.4 - 433.8	404.4	431.4	458.4	0.16%	1 \pm 0.53%

Both complexes have a measured polydispersity of 1, indicating that the peak represents a **single monodisperse species** instead of a mix of species with different stoichiometries. Furthermore, in both cases, the closest mass aligns with a ratio of two CDAN1, three ASF1A (each ~26 kDa with the epitope tag), and for C:C:A, two CDIN1, completely consistent with the mass difference between the C:C:A and C:A complex. Because SEC-MALS was performed on complexes purified independently from the prep used for cryo-EM, these are **orthogonal validations** of the predominant ratio of two CDAN1 and three ASF1A in our samples. In addition, we now note in the text that although these observations support the predominant stoichiometry of our recombinant complexes, endogenous complexes may have more variable stoichiometries (page 10, lines 28-30).

I also wonder why the 4th copy of Asf1 would not bind to the second CDAN1?

We can only speculate why a 4th copy of ASF1 does not favorably or stably bind. One possibility is that a 4th copy could bind at higher ASF1 concentrations. However, the high expression levels achieved with transfection combined with ~0.5 μ M endogenous ASF1 (opencell.czibiohub.org) argue against this possibility. A second possibility may be steric hindrance from unresolved regions of CDAN1 that either asymmetrically bind the resolved regions of the complex or become restrained upon binding of the 3rd ASF1 molecule. A third related possibility is that the binding of one distal ASF1 prohibits conformational changes required to stably engage a 4th ASF1. Both motifs (BD_{A2} and HMH_{A2}) that bind the distal ASF1 are located in a disordered loop in MIF4G1 of CDAN1-1, and stable stacking of one distal ASF1 may restrict the flexibility of nearby regions. Consistent with some of these ideas, 3D variability analysis of our cryo-EM data (new **Supplementary Movie 1**) shows:

1. Evidence for conformational heterogeneity in the complex.
2. Asymmetry in all conformations.
3. Additional low-resolution extra densities, including above ASF1A-3 in some conformations. However, this density is only visible at low map thresholds and does not resolve clearly in cryo-EM maps after high-resolution refinements. This low-resolution density above ASF1A-3 does not obviously fit another ASF1 molecule stacked in the same configuration we see in our high-resolution maps and is not consistently present when two ASF1 are stably stacked on the other side of the CDAN1 dimer.

These observations support a dynamic process for stably engaging and stacking two ASF1; if rearrangements in both copies of CDAN1 are required to 'stack' a distal ASF1, then these motions may be prevented upon binding of one distal ASF1.

2) Fig. 3 was very helpful to understand authors' structure, however, other figures (Fig. 4 Fig. S7d, Fig. S8b etc.) are likely viewed from different orientations. Clearer connections between these figures would greatly enhance their utility in conveying the structural relationships.

We have updated structural figure panels throughout the manuscript to show clearer relations with the overall structure.

3) Figure S7 contains critical information but may overwhelm general readers. Especially it would be helpful to clarify which regions are included in the final deposited structure.

We have updated **Extended Data Fig. 7** to clarify the information presented. We specify that the final deposited structural model includes HMH_{A1}, BD_{A1}, and BD_{A2}, but not HMH_{A2}.

4) The authors have not deposited the CDAN1-CDIN1 structure in EMDB or PDB, making it difficult for readers to assess or even view the structure. Furthermore, the manuscript lacks details on the statistics and refinement procedures. While 6 Å resolution should allow for visualization of α -helices, Figure S6 does not offer sufficient clarity for such assessments.

We added details regarding the cryo-EM processing procedure for the map of the CDAN1-CDIN1 complex to **Extended Data Fig. 6** and the Methods section. We have also uploaded the cryo-EM map and Alphafold3 prediction into the figshare repository associated with this manuscript (<https://doi.org/10.6084/m9.figshare.26414005>). However, we chose not to deposit these files into the EMDB and PDB because we did not use the cryo-EM map for any structural modeling besides rigid body docking of an Alphafold3 prediction. The low resolution and preferential orientation present in the map (which may result in overestimation of the resolution) prohibit further interpretation.

Other minor comments are as follows:

P5: This domain architecture, particularly the distinctive presence of the three-helix bundle, resembles the central portion of the CNOT1 protein that serves as a scaffold in the CCR4-NOT deadenylation complex^{36,45–48} (Extended Data Fig. 5c-f). This observation is intriguing, but it would benefit from being included in the Discussion section, where it could be explored in greater depth.

We added additional discussion of this similarity (page 11, lines 20) as follows:

“Future insights into CDAN1 complexes will require additional studies of CDIN1 and other domains of CDAN1. Notably, structural homology searches revealed that CDAN1 exhibits extensive homology to CNOT1 (**Extended Data Fig. 5**), the scaffolding component of the CCR4-NOT deadenylation complex^{36,45–49}. While human CNOT1 (2,376 residues) is larger than CDAN1 (1,227 residues), an internal ~1,000 residue segment of CNOT1 folds into a series of four domains that are structurally homologous or similar to the entirety of CDAN1. These four domains are also encoded in the same order in both proteins. Structural homology searches of the two MIF4G domains and the DUF3819 three-helix bundle of CDAN1 in isolation against the predicted human proteome consistently identify the corresponding domain in CNOT1 as a top match⁴⁵. The fourth domain a member of the HEAT family that forms an MIF4G-like domain in CNOT1 but more closely resembles an MA3 domain in CDAN1. MIF4G domains are commonly found in translational regulators, often in tandem with other shared domains, such as MA3^{44,47,55}. However, no other MIF4G proteins contain predicted homology to DUF3819⁴⁵. These findings suggest that CDAN1 and CNOT1 may have evolved from a common ancestor and have related functions. The possibility that CDAN1 controls substrate access for the putative nuclease function of CDIN1, as an analogy to the CNOT1-mediated coordination of the Ccr4/Caf1 nucleases during deadenylation, is an intriguing comparison for future investigation.”

P6: BDA2 also extends into additional conserved residues that wrap around ASF1A-2, which may help stabilize the position of ASF1A-2 stacked on top of ASF1A-1. BDA2 is apparently much longer than the canonical B-domain. Why not show key interactions between the extended C-terminal region and Asf1 in Fig. 4B?

We have updated the text to denote this region as a canonical B domain (e.g., BD_{A2}) with a C-terminal extension and added information regarding potential interactions made by this extension in **Extended Data Fig. 7d, 8c, and 9a**. However, the resolution of this region is lower than other parts of the complex. Thus, although we can trace the general path of the extension

(new **Extended Data Fig. 4d**) and the sequence is highly conserved, we cannot confidently assign residue-specific interactions.

Also, as far as I know, the canonical B-domain is always stabilized by forming a beta-sheet. Why not refine the structure by applying some constraints to enforce the hydrogen-bonding between beta-strands. It is much easier for readers to find the B-domain if we see its corresponding b-strand arrow.

We agree; this is now reflected in the updated model and figures.

P7: Specifically, a loop (residues 469-476) associated with CDAN1 MIF4G2 would clash with an H4 tail positioned on ASF1A-1 and the extended BDA2 loop would clash with an H4 tail positioned on ASF1A-2 (Extended Data Fig. 7d). It would benefit by relating this to the entire structure. Also, for the left panel of Fig. S7d, some constraints to enforce b-strands of Asf1 should be applied during refinement.

We have updated **Extended Data Fig. 7d** to make these points clearer.

P8: The histone chaperone HIRA has also been reported to bind ASF1A with higher affinity, in part by distinguishing the variable NTD of ASF1A and ASF1B that lie proximal to the Bdomain interface²⁸ (Extended Data Fig. 8b). The reference 28 has shown that HIRA does not distinguish between ASF1a and ASF1b through the interaction between the HIRA B-domain and the ASF1 core domain.

We apologize for any confusion. We meant to describe that HIRA does **not** use its B-domain to distinguish between ASF1A and ASF1B and instead uses a region proximal to the B domain. We have rewritten the text (page 9, lines 15-17) to clarify this point.

This biochemical analysis is intriguing, but it would be more informative if the authors clearly show which residues of Asf1a, but not Asf1b, are critical for binding.

The N-terminal swaps span 27 amino acids but only 8 amino acids differ in this region, as indicated in the sequence alignment (**Extended Data Fig. 8b**) and below:

Residue #	5	7	10	13	19	20	23	27
ASF1A	Q	N	V	D	Y	N	Q	T
ASF1B	S	L	A	E	H	S	R	S

Thus, the swaps are essentially 8 point mutations. We also tested specific mutations of ASF1B residues to the corresponding amino acid in ASF1A – in particular, 1) R23Q or 2) S5Q/ L7N combined, to see if they could rescue binding to HMH_{A1}* ST-CDAN1. These mutations significantly change the biochemical properties of the amino acids and map to regions of ASF1A most likely to influence stable interaction with CDAN1. Residue 23 of both the proximal and distal ASF1A faces the interface between the two ASF1A molecules in the stack, close to the orthogonal B-domain that interacts canonically with the other ASF1A molecule (**Extended Data Fig. 8c**). Notably an arginine in this position would be in close proximity to a basic residue on the orthogonal BD (new **Extended Data Fig. 9a**). However, R23Q ASF1B did not enhance ASF1B binding to HMH_{A1}* ST-CDAN1 (new **Extended Data Fig. 9b**). We also examined S5Q/L7N ASF1B. On the distal ASF1, these two residues are close to the backbone of the conserved extension following BD_{A2}. S5Q/L7N ASF1B partially rescued binding to HMH_{A1}* ST-CDAN1, but not to HMH_{A1}* BD_{A1}* ST-CDAN1 (new **Extended Data Fig. 9c**; compare with **Fig. 5b**, bottom). This partial effect suggests that multiple residues in the NTD contribute to the differential recognition of ASF1 paralogs by CDAN1.

The side chains are difficult to discern in Figure S8. Additionally, it's unclear why BDA1 and BDA2 are positioned differently between the left and right panels. Readers would benefit from clearer guidance on these points.

We adjusted **Extended Data Fig. 8c** to more clearly show the variant residues on each ASF1 molecule, aligned to show the same view of ASF1 in each case. We also added **Extended Data Fig. 9a** showing the NTD on each ASF1 relative to each CDAN1 binding element in the stack.

Reviewer #2

This study presents interesting and novel findings about the mechanism by which CDAN1 could inhibit ASF1 histone chaperone activity. The structure of the CDAN1 complex with CDIN1 and ASF1 supports a 1:1:>1 stoichiometry and reveals novel ASF1 binding regions in CDAN1, which would be expected to be competitive with histone H4 and H4. The results provide generally adequate support for the authors conclusions, except for the title. Specifically, the manuscript could be strengthened with more complete mutagenesis studies and competition experiments.

We thank the reviewer for their feedback. In addition to the point-by-point replies below, we note that we have changed the manuscript title to “Mechanism of ASF1 Engagement by CDAN1”

1) A major conclusion is that CDAN1 can bind to 2 ASF1 molecules using 2 B domain (BDA2) and H3- helix 3 mimics (HMHA1 and HMHA2). To this reviewer, the cryo-EM is not completely clear about these interactions as the density seems to be weak and the placement of the binding domains is largely through modeling (Extended data figures). Given this possibility for ambiguity in the placement of the binding motifs, further structure validation is needed. The mutagenesis studies only addressed BDA2 and HMHA1 and would benefit from also testing combinations of mutants of BDA1 and HMHA2 and the use of disruptive mutations in addition to alanine.

We added mutagenesis analyses validating that **all four binding elements we identify on CDAN1 contribute to ASF1 binding** (new **Fig. 5** and **Extended Data Fig. 8a**). We note that these are point mutations of only three residues in each binding element to alanines, which we believe is a highly specific strategy. We do not immediately see the added value of “disruptive mutations in addition to alanine” after observing clear and specific loss-of-function effects with alanine point mutants. We also note that the map densities for $HMHA_1$, BD_{A1} , and BD_{A2} are not weak: all three elements are clearly visible without needing to adjust the threshold of the cryo-EM maps (**Fig. 3b,c** and **Extended Data Fig. 4d**). Only visualizing $HMHA_2$ requires additional adjustments (**Extended Data Fig. 7c**), likely because it is more flexible as the element furthest away from the structured dimer core. Therefore, it is clear that the $HMHA_1$, BD_{A1} , and BD_{A2} interactions with ASF1 exist. Instead, although modeling $HMHA_1$ was unambiguous, the major challenge we faced was distinguishing the precise sequence for each B-domain density. Both BD_{A1} and BD_{A2} are short, and the sidechains of the residues in them are very similar (**Fig. 4d**). Thus, although the presence of the two B-domain densities was clear, the resolution of the map in those areas was not sufficient to confidently distinguish a lysine from an arginine or a threonine from an asparagine. As is standard in such circumstances, we turned to additional information to interpret the structure:

1. While BD_{A2} was already annotated as a B-domain, BD_{A1} was not. However, it clearly has the sequence properties of a B-domain (**Fig. 4d**) and extends directly from the folded region of CDAN1 that interacts with the proximal ASF1A-1. Notably, we have now validated both B-domain sequences through sufficiency experiments (see below) and mutagenesis, with the notable result that mutating both BD_{A1} and BD_{A2} , but not either individually, abrogates ASF1 binding (**Fig. 5a**), supporting their nonredundant functional roles.

2. To determine which B-domain sequence corresponded to each B-domain density, we noted:
 - a. The BD_{A1} sequence immediately follows the CDAN1 domain that interacts with ASF1A-1. Importantly, **this linker is too short for the BD_{A1} sequence to reach the density corresponding to BD_{A2}** (new **Extended Data Fig. 7a**).
 - b. In contrast, the BD_{A2} sequence is on a long loop that can reach the distal ASF1A-2, and the conserved residues C-terminal of BD_{A2} (but not BD_{A1}) can explain the additional density observed in our maps that extends from the BD_{A2} density and wraps around ASF1A-2 (**Fig. 3 and new Extended Data Fig. 4d**).
3. The interpretations above are also fully supported by Alphafold3 predictions (**Extended Data Fig. 7b**). Thus, although we use Alphafold to help interpret our cryo-EM maps, the predictions are only one among numerous considerations; only the assignment of HMH_{A2} is strongly reliant on Alphafold prediction. For this reason, we do not include HMH_{A2} in our atomic model (**Extended Data Fig. 7c**) but discuss it in the text and through figure panels showing the Alphafold prediction (**Fig. 4g and Extended Data Fig. 7b-d**). In all cases, the Alphafold3 prediction is clearly noted in the figure itself or in the legend.

Finally, we demonstrate using ITC that isolated GST-tagged B-domains, GST-BD_{A1} and GST-BD_{A2}, are sufficient for binding recombinant ASF1A, whereas GST-BD_{A1}^{*} and GST-BD_{A2}^{*} carrying mutations as in **Fig. 5a** exhibit no detectable interaction (**Reviewer Fig. 1**). Because ASF1A binding to these individual elements is specific but technical considerations (including a weak affinity to BD_{A1}) prevent reliable estimates of K_d values, we do not include these data in the manuscript. Importantly, our mutagenesis studies already indicate that avidity through multiple elements plays an important role in ASF1 engagement by CDAN1.

Reviewer Fig. 1. Isothermal titration calorimetry (ITC) of ASF1A and CDAN1 B-domains. GST fused to BD_{A1} (residues 818-841), BD_{A1}^{*} (BD_{A1} M824A/R825A/K826A), BD_{A2} (CDAN1 residues 183-218), or BD_{A2}^{*} (BD_{A2} S194A/R195A/R196A), and FLAG-ASF1A were purified from *E. coli*. ITC with a Microcal ITC200 was performed with 50 μM ASF1A loaded in the cell and GST fusions loaded in the syringe at 500 μM. GST fusions were titrated in at 25°C with 19 injections (2 μL each). Normalized readings are displayed. ASF1A interaction is detected above baseline with BD_{A1} (teal) or BD_{A2} (blue) but not with either mutant (BD_{A1}^{*} or BD_{A2}^{*}, grays).

Altogether, while we note that *all* structural models deposited to the PDB are only interpretations of structural data (in this case, cryo-EM maps), we believe multiple orthogonal lines of evidence support the assignments of BD_{A1}, BD_{A2}, and HMH_{A1} in our model, and that the evidence supporting the role of HMH_{A2} is worth discussing.

2) The possible mechanism of inhibition of ASF1 function is interesting, but there isn't strong direct evidence that CDAN1 actually inhibits histone binding to ASF1. The binding affinity of ASF1 for histones is very strong and this study does not test whether CDAN1 can compete with

histones for ASF1. Further studies including binding affinities and competition experiments would be needed to prove this statement, which is the title of the manuscript.

We agree with the reviewer that our implication that CDAN1 inhibits ASF1 is derived primarily from the observation that CDAN1 binding to ASF1 occupies all known functional binding sites (e.g., the B-domain binding site and the histone binding site) of ASF1. Considered together with the cytosolic localization of CDAN1, ASF1 bound to CDAN1 is unlikely to carry out any of its canonical functions in nuclear shuttling and assembly of H3-H4. The Jeong*, Frater*, et al. preprint (doi: 10.1101/2024.07.10.602876) also made a claim of inhibition and showed that excess CDAN1 can prevent with H3 binding to ASF1 but cannot compete H3 off of ASF1. We observed a similar phenomenon at very high concentrations of CDAN1 (**Reviewer Fig. 2**), albeit at ratios of these factors unlikely to be present in most cells. We also tried using ITC to assess the sufficiency of ASF1A binding to GST-tagged histone H3 helix (GST-H3H), GST-HMH_{A1}, and GST-HMH_{A2}, along with corresponding mutants. However, although we could detect specific ASF1A binding with the B-domains using ITC (**Reviewer Fig. 1**), we did not obtain clean measurements for GST-H3H and ASF1A, a positive control needed to compare affinities of ASF1A with the H3 helix and HMH variants. With these considerations, we have changed the title and toned down the implication of direct inhibition in the manuscript.

3) The observation that the BD and HMH interactions with the multiple ASF1 proteins is not symmetric and that the stoichiometry is not also symmetric (1:1:2 or 2:2:4) is very surprising. SEC was the only method used to support the unusual stoichiometry (2:2:3) and this has error associated with it. Please explain how any of the EM approaches (grid preparation – complex falling apart; or averaging, etc.) might have influenced the appearance of the complex. Is there a better explanation for this unusual stoichiometry? Equilibrium analytical ultracentrifugation would provide a more accurate estimate of the size of the complex.

We refer to our response to **Reviewer #1's first point** for detailed descriptions. Briefly, both SEC-MALS (not only SEC) measurements of molar mass and consistent observations from our cryo-EM data processing **orthogonally support** the 2:2:3 CDAN1:CDIN1:ASF1A stoichiometry and are inconsistent with 2:2:2 or 2:2:4 stoichiometries. Thus, while there may be technicalities involved with each approach, there is strong support for the stoichiometry of our recombinant complexes. In addition, we now note that endogenous complexes may have more variable stoichiometries (page 10, lines 28-30) and the presence of conformational heterogeneity (new **Supplementary Movie 1**). We do not have ready access to resources to perform equilibrium analytical ultracentrifugation but note that SEC-MALS is a widely accepted technique to measure macromolecular mass in solution and for analyzing stoichiometry. Finally, while we

cannot be certain what causes this stoichiometry, we speculate that binding of the 3rd ASF1 may prevent conformational changes required to stably engage and ‘stack’ the 4th ASF1.

4) There are results derived from only two (not 3 or more) independent replicates.

All experiments presented in the manuscript have now been replicated 3 or more times.

5) Figure 2: what are the doublets in Fig. 2a and b for F-CDIN1, HA-ASF1A and F-ASF1A?

We do not know exactly what causes the doublet bands but note they are consistently observed with overexpression of tagged ASF1 in eukaryotic cells – e.g., Supplementary Fig. 3b in Campos et al. (PMID: 20953179) and Fig. 5b,d in Sanematsu et al. (PMID: 16537536). The additional band is assumed to be a posttranslational modification. Consistent with this interpretation, we do not observe doublets of HA-ASF1A purified from *E. coli* (**Reviewer Fig. 3a**), suggesting a eukaryotic-specific modification rather than altered SDS-PAGE migration patterns due to protein characteristics or epitope tag. We believe a similar reasoning applies to CDIN1 (**Reviewer Fig. 3b**). Although the exact modification is not certain, both ASF1A and CDIN1 are reported to be phosphorylated in human cells (PMID: 16537536, 23312004, 35136069, 24598821) and *in vitro* phosphorylation of ASF1A purified from *E. coli* with TLK2 indeed causes a size shift (**Reviewer Fig. 3c**).

Reviewer Fig. 3. Investigation of doublet bands.

a, SDS-PAGE and Coomassie staining shows that recombinant GST-tagged HA-ASF1A purified from *E. coli* after GST cleavage with GST-3C protease (left) and subtraction using Glutathione Sepharose (middle) does not run as a doublet. **b**, *In vitro* phosphorylation assay of 60 μ M HA-ASF1A purified as in **a** with 5 μ M GST-TLK2, a kinase that phosphorylates ASF1 without or with 1 mM ATP at 32°C for 30 minutes, analyzed by SDS-PAGE and Coomassie staining. Note: ATP-dependent phosphorylation results in a shift of the HA-ASF1A to a higher molecular weight species (pHA-ASF1A). *, contaminant from the GST-TLK2 purification. **c**, FLAG-CDIN1 (F-CDIN1) purified from Expi293F cells (left), which runs as a doublet or *E. coli* (right), which runs as a single band.